# Deep learning based sub-seasonal precipitation and streamflow ensemble forecasting over the source region of the Yangtze River

Ningpeng Dong[1], Haoran Hao[2,1], Mingxiang Yang[1], Jianhui Wei[3], Shiqin Xu[4], Harald Kunstmann[3,5,6]

[1]State Key Laboratory of Simulation and Regulation of Water Cycle in River Basin, China Institute of Water Resources and Hydropower Research, Beijing, China
[2]State Key Laboratory of Hydraulic Engineering Intelligent Construction and Operation, Tianjin University, Tianjin, China
[3]Institute of Meteorology and Climate Research (IMKIFU), Karlsruhe Institute of Technology, Campus Alpine, Garmisch-Partenkirchen, Germany
[4]Hydrology, Agriculture and Land Observation (HALO) Laboratory, King Abdullah University of Science and Technology, Thuwal, Saudi Arabia
[5]Institute of Geography, University of Augsburg, Augsburg, Germany
[6]Centre for Climate Resilience, University of Augsburg, Augsburg, Germany

*Correspondence to*: N. Dong (dongnp@iwhr.com) and H. Hao (1020205041@tju.edu.cn)

**Abstract.** Hydrometeorological forecasting is crucial for managing water resources and mitigating the impacts of extreme hydrologic events. At sub-seasonal scales, readily available hydrometeorological forecast products often exhibit large uncertainties and insufficient accuracies to support decision making. We propose a deep learning based modelling framework for sub-seasonal joint precipitation and streamflow ensemble forecasts for a lead time of up to 30 days. This is achieved by coupling (1) an ensemble of enhanced convolutional neural network (CNN) models with ResNet blocks and a specialized loss function for statistically downscaling of ECMWF ensemble precipitation forecasts to (2) a hybrid hydrologic model integrating the conceptual Xin'anjiang model (XAJ) and the long-short term memory network (LSTM) for ensemble streamflow forecasting. Applying the modeling framework to the source region of the Yangtze River Basin, results indicate that the CNN-based downscaling model exhibits ~34% and ~26% less RMSE than the raw ECMWF forecasts and the quantile mapping (QM) forecasts, respectively, averaged over the 30-day lead time. Similarly, the CNN achieves approximately 6% and 10% lower RMSE than raw forecasts and QM for heavy precipitation events. Using these precipitation forecasts as meteorological drivers for the hybrid XAJ-LSTM hydrologic model, we found that forecasted streamflow and flood peaks driven by CNN-based precipitation forecasts have 16%-33% lower relative errors and 20%-31% lower RMSE compared to those driven by raw forecasts. However, the standalone XAJ model shows only marginal improvements with the same enhanced precipitation forecasts. This highlights the importance of understanding the effectiveness of the hydrologic model as part of the sub-seasonal hydrometeorological modeling chain. Our study is expected to provide implications for leveraging advanced AI techniques to enhance sub-seasonal hydrometeorological forecasting accuracy and operational efficiency for effective water resources management and disaster preparedness.

## 1 Introduction

In past decades, the frequency and intensity of extreme precipitation events are increasing in many areas as global warming continues, thereby amplifying the potential for hazards of extreme weather and hydrologic events (Wei et al., 2018; Yuan et al., 2018; Wang et al., 2019; Zhu et al., 2020). Hydrological forecasting has become critically important for managing water resources and mitigating the impacts of these extreme weather and hydrologic events (Robertson and Wang, 2013; Liu et al., 2020; Jiang et al., 2022). Traditional hydrological forecasts, which do not integrate sub-seasonal meteorological forecasts, often provide insufficient lead times for decision-making on flood control, agricultural planning, and ecological preservation efforts (de Andrade et al., 2021; Bierkens, 2015; Jaun et al., 2008). Integrating both meteorological and hydrological forecasts at sub-seasonal scales is therefore essential to extend lead times, thereby improving water resources management and disaster preparedness over a longer term (Yuan et al., 2016; Cloke and Pappenberger, 2019; Liang et al., 2018; Vigaud et al., 2019; Zhu et al., 2019).

Advancements in numerical weather prediction (NWP) models, such as the ECMWF Integrated Forecasting System (IFS) and the NCEP Global Forecast System (GFS), have greatly improved the accuracy of sub-seasonal weather forecasting (Yuan et al., 2011; Bauer et al., 2015; Brotzge et al., 2023). However, these global models often suffer from relatively coarse resolutions and generalized parameterizations that may not be suitable for regional-scale and local-scale forecasts (Dehshiri et al., 2023; Singhal et al., 2023). Dynamic downscaling, such as that performed by the Weather Research and Forecasting (WRF) model, translates larger-scale atmospheric trends captured by GCMs into fine-scale regional details that reflect local geographic and climatic factors (Merino et al., 2022; Nooni et al., 2022; Maraun et al., 2010). For instance, recent studies by Gao et al. (2022) and Srivastava et al. (2023) demonstrate the effectiveness of WRF in enhancing the accuracy of precipitation forecasts and capturing the dynamics of severe weather events. Despite these advantages, dynamic downscaling often requires extensive computational resources especially for sub-seasonal scales, and can be sensitive to the quality of input data. Furthermore, the process is constrained by the physical parameterizations that may not always accurately represent localized meteorological conditions, a concern that is increasingly critical under changing climatic conditions (Di Luca et al., 2015; Shi, 2020; Xu et al., 2015).

Statistical downscaling techniques, which have been used to relate the larger-scale meteorological patterns to local-scale weather, offer a different approach (Tabari et al., 2021; Zhang et al., 2022; Michalek et al., 2024). Traditional statistical downscaling methods such as quantile mapping have proven effective in reducing the systematic bias of precipitation forecasts with relatively simple inputs (Wilby et al., 2004; Vrac & Friederichs, 2015). On the other hand, forecasting weather and predicting climate using machine learning, especially deep learning (DL), has recently become a hot topic. A common approach for this purpose is to use preceding predictors from observational or reanalysis data to forecast subsequent predictands (Weyn et al., 2021; Xie et al., 2023; Ham et al., 2019; Kalnay et al., 1996; Ling et al., 2022). An alternative method involves postprocessing dynamical forecasts. For instance, Cho et al. (2020) applied machine learning techniques, including random

forests and support vector machines, to develop statistical relationships for temperature adjustments. Similarly, Kim et al. (2021) utilized Long Short-Term Memory (LSTM) networks to correct bias in the amplitude and phase of the Madden–Julian Oscillation. More recently, deep learning models such as convolutional neural networks (CNN) have been reported able to more effectively reduce the total bias of meteorological forecasts due to their ability to learn multi-dimensional representations of data features (Vandal et al., 2019; Sachindra et al., 2018; Jiang et al., 2024; Li et al., 2022). For example, Lagerquist et al.

(2019) used a CNN to identify fronts in gridded data for spatially explicit prediction of synoptic-scale fronts.

Despite general improvements of forecasts, these DL-based models tend to smooth the extreme precipitation at sub-seasonal scales (Baño-Medina et al., 2021; Kim et al., 2022), likely due to insufficient heavy precipitation samples (Chen et al., 2022). Many studies have since introduced more recent variants of CNNs including the U-shaped U-Net (Han et al, 2021; Horat and Lerch, 2024; Ni et al., 2023) and SmaAt-UNet (Li et al., 2024), or coupled standard CNNs with different structures, such as

Auto-Encoder (Ling et al., 2022) and Transformer (Ling et al., 2024). In particular, the residual network, ResNet, has been introduced in sub-seasonal forecast correction, which shows the potential of mitigating the vanishing gradient issue by introducing the residual paths (Jin et al 2022; Nie et al., 2024). Others have attempted to introduce specialized loss functions to balance heavy and light rains, such as the exponentially weighted mean squared error (Ebert-Uphoff et al., 2020) and Dice loss (You et al., 2022). However, these new developments have not been sufficiently examined for sub-seasonal forecasts.

Other state-of-the-art forms of deep learning for weather forecasts include fully DL-based models, such as Pangu (Bi et al., 2023) and GraphCast (Lam et al., 2023), which are reported able to achieve forecast skills comparable to numerical weather prediction systems. While these models may appear quite different from statistical post-processing deep learning models, some argue that these models act more as post-processing tools rather than realistic simulators of the atmosphere due to the lack of physical fidelity and consistency (Bonavita et al., 2024). Although not the primary focus of this paper, this calls attention to

the scientific community to critically evaluate and differentiate between the capabilities and applications of fully DL-based models and DL models designed for post-processing.

In addition to meteorological forecasts, sub-seasonal streamflow forecasts are crucial because streamflow at these timescales is directly related to the onset and progression of flooding and drought events. To translate meteorological predictions to streamflow forecasts, both physics-based and data-driven hydrologic forecasting models are widely used. Physics-based

models, such as the lumped Xin'anjiang model, HBV model, and the distributed CLHMS and VIC models, make predictions by interpreting detailed physical processes (Gassman et al., 2014; Arnold et al., 2015; Dong et al., 2022; Dong et al., 2023). Data-driven models were also developed to perform rainfall-runoff modelling and forecasts by learning from big data (Kisi, 2007; Adnan et al., 2019), and have been reported to outperform the well-calibrated physics-based models (Kratzert et al., 2019). It is noteworthy that both models are embedded with uncertainties. Physics-based models may produce inaccurate

simulations due to simplified representations of hydrologic processes, and data-driven models may perform less effectively in extrapolation beyond the range of the training data (Addor et al., 2020). By integrating the strengths of both approaches, recent

studies have attempted to establish a hybrid model with a higher predictive performance than the physical model alone (Liu et al., 2022; Abrahart et al.,2012; Raftery et al.,2005; Yang et al.,2020). For example, Humphrey et al. (2016) combined a Bayesian neural network (BNN) with the traditional GR4J model and achieved improved forecast accuracy compared to using either the BNN or GR4J alone. However, the role of such models as part of the hydrometeorological modelling chain in producing reliable streamflow forecasts has not been well examined at sub-seasonal scales. For example, Crochemore et al. (2016) and Valdez et al. (2022) suggested that the relationship between the accuracy of precipitation forecasts and the corresponding streamflow forecasts is not necessarily straightforward. Ensemble approach with DL models also shows promising results (Ferranti et al., 2018; Bremnes, 2020; Balint et al., 2006; Cloke and Pappenberger, 2009; Scheuerer and Hamill, 2015; Taillardat et al., 2016) that require further investigation.

The above considerations are particularly relevant for the source region (SR) of the Yangtze River Basin, which is historically susceptible to extensive flooding and droughts that affect thousands of kilometers downstream (Sun et al., 2016). Aiming at enhancing sub-seasonal hydrometeorological forecasts for the wet season in this area, we start by addressing the following questions:

(1) How effectively can CNN architectures with recent extensions improve the sub-seasonal precipitation forecasts compared to traditional statistically downscaling models?

(2) How effectively can AI-assisted hydrologic models convert more accurate sub-seasonal precipitation forecasts into more accurate streamflow forecasts compared to traditional conceptual hydrologic models?

Specifically, this study investigates sub-seasonal precipitation and streamflow ensemble forecast skills for up to 30 days ahead with deep learning models, which integrates enhanced CNN models with ResNet blocks and specialized loss functions for post-processing the ensemble ECMWF forecasts with a hybrid hydrologic model of the Xin'anjiang model (XAJ) and the long-short term memory network (LSTM) for streamflow forecasting. Our approaches and findings are expected to provide implications for operational hydrometeorological forecasts in the SR and also similar basins worldwide.

## 2 Study Area and Data

### 2.1 Study Area

The source region (SR) of the Yangtze River basin is located on the eastern edge of the Tibetan Plateau, between 90°E to 101°E and 26°N to 36°N, as shown in Figure 1. The region serves as a crucial transitional area from highland mountains to plains in southern China, with the surface elevation decreasing from over 6000 m in the north to just over 2500 m in the south. The climate of the region is subject to both plateau and subtropical monsoon climates, with annual precipitation ranging from 280 mm to 760 mm. The SR has a significant impact on the utilization of water resources in the Yangtze River Basin and Southwest China (Hao et al., 2024). The controlling hydrologic station of the SR is Shigu station (Fig. 1), which has a mean annual streamflow of around 1300 $m^3$/s, accounting for 5% of the total water resources of the Yangtze River Basin.

The SR spans a large north-south range, with the northern region deep in the Tibetan Plateau and dominated by a plateau monsoon climate, and the southern region characterized by low hills and a subtropical monsoon climate. These contrasting environments suggest different runoff generation mechanisms between the two regions. To ensure the accuracy of streamflow simulations at Shigu, we therefore divided the SR into two sub-basins (i.e., northern and southern basins in Figure 1) for hydrologic modeling in Section 3.4.

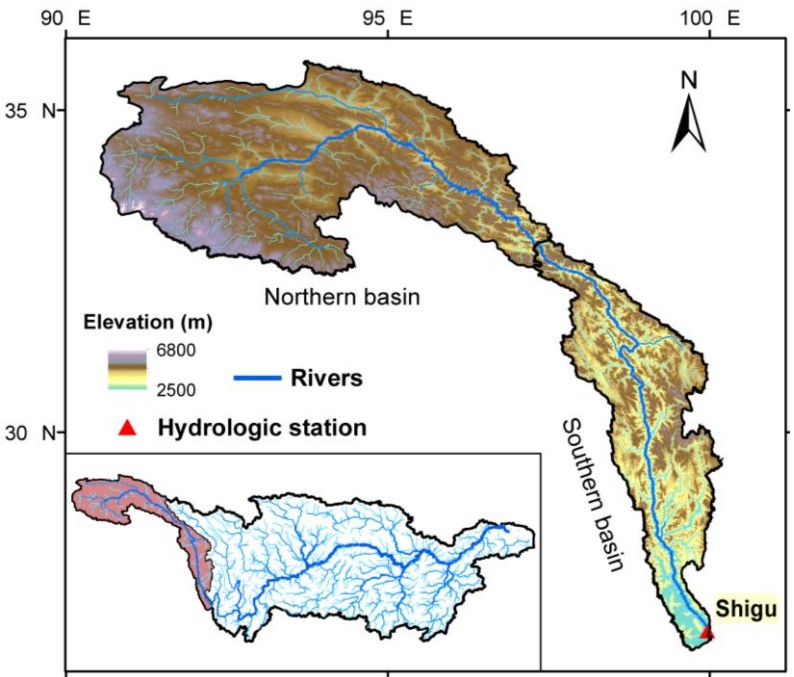

**Figure 1. The source region of the Yangtze River Basin and its location in the Yangtze River Basin.**

**2.2 Data sources**

*Observed precipitation and temperature.* To train the forecast models and evaluate the accuracy of forecasts, this study employs the 0.25° daily precipitation and temperature grid dataset (CN05.1), released by the National Meteorological Information Center, as the reference observed data. This dataset is produced by interpolating precipitation and temperature data from over 2,000 meteorological stations across the country and covers the period from 1961 to 2022.

*ECMWF sub-seasonal reforecast data.* The European Centre for Medium-Range Weather Forecasts (ECMWF) offers a sub-seasonal forecast service designed to bridge the gap between short-range weather predictions and long-term climate outlooks. This service focuses on predicting atmospheric and oceanic conditions over the next 2 to 6 weeks, providing valuable information for a variety of applications such as water resources management, and disaster preparedness. In this study, we collect the ECMWF Sub-seasonal to Seasonal (S2S) daily reforecast data for a lead time of 30 days initialized on 35 dates during the wet season (between May and August) per year during 2002-2019. The forecasted variables used in this study

include precipitation, convective precipitation at the land surface, and temperature, wind components, geopotential heights, and specific humidity at 200/500/850hPa pressure levels. All of these variables are at a spatial resolution of 1.5°.

*Observed streamflow.* To calibrate the hybrid hydrologic model and evaluate the hydrologic forecasts, the daily streamflow data of Shigu station is collected for 1981-2019.

## 3 Methods

### 3.1 Overview

The presented sub-seasonal hydrometeorological forecasting framework aims to improve the daily precipitation forecasts and the corresponding streamflow forecasts at the Shigu hydrologic station during the wet season (May to August) for a lead time up to 30 days. We first employ all 10 ensemble members from the ECMWF S2S gridded sub-seasonal precipitation reforecast dataset for the next 30 days as raw forecasts, denoted as EC. An ensemble of enhanced CNN models with ResNet blocks and a specialized loss function is established to statistically downscale and bias correct each ensemble member of the 1.5° EC raw precipitation forecasts to 0.25° grid resolution, with its post-processed forecast denoted EC-CNN (Section 3.2.1). The quantile mapping (QM) serves as a benchmark for comparison, with its post-processed forecast denoted EC-QM (Section 3.2.2).

These three gridded ensemble precipitation forecasts (EC, EC-QM and EC-CNN), along with ECMWF gridded sub-seasonal daily temperature forecasts corrected by the delta method (Section 3.3), are employed to drive two lumped hydrologic models to produce the daily streamflow forecasts for lead times of 1-30 days. All these gridded forecasts are areal-averaged over the two sub-basins of the SR (Figure 1) before being input to the lumped hydrologic models. The first hydrologic model is a standalone XAJ model (Section 3.4.1), and the second model is a hybrid model that integrates the conceptual XAJ model and the LSTM (hereinafter XAJ-LSTM) (Sections 3.4.2 and 3.4.3). The streamflow and flood forecasts of XAJ-LSTM and standalone XAJ driven by EC-CNN forecasts are then quantitatively evaluated against those driven by EC and EC-QM forecasts.

The evaluation metrics include deterministic metrics of Root Mean Squared Error (RMSE), Relative Error (RE) and the Nash-Sutcliffe efficiency (NSE) for the ensemble mean, and probabilistic metrics of Continuous Ranked Probability Score (CRPS) for a total of 10 ensemble members (Section 3.5). A detailed workflow of this study is presented in Fig. 2.

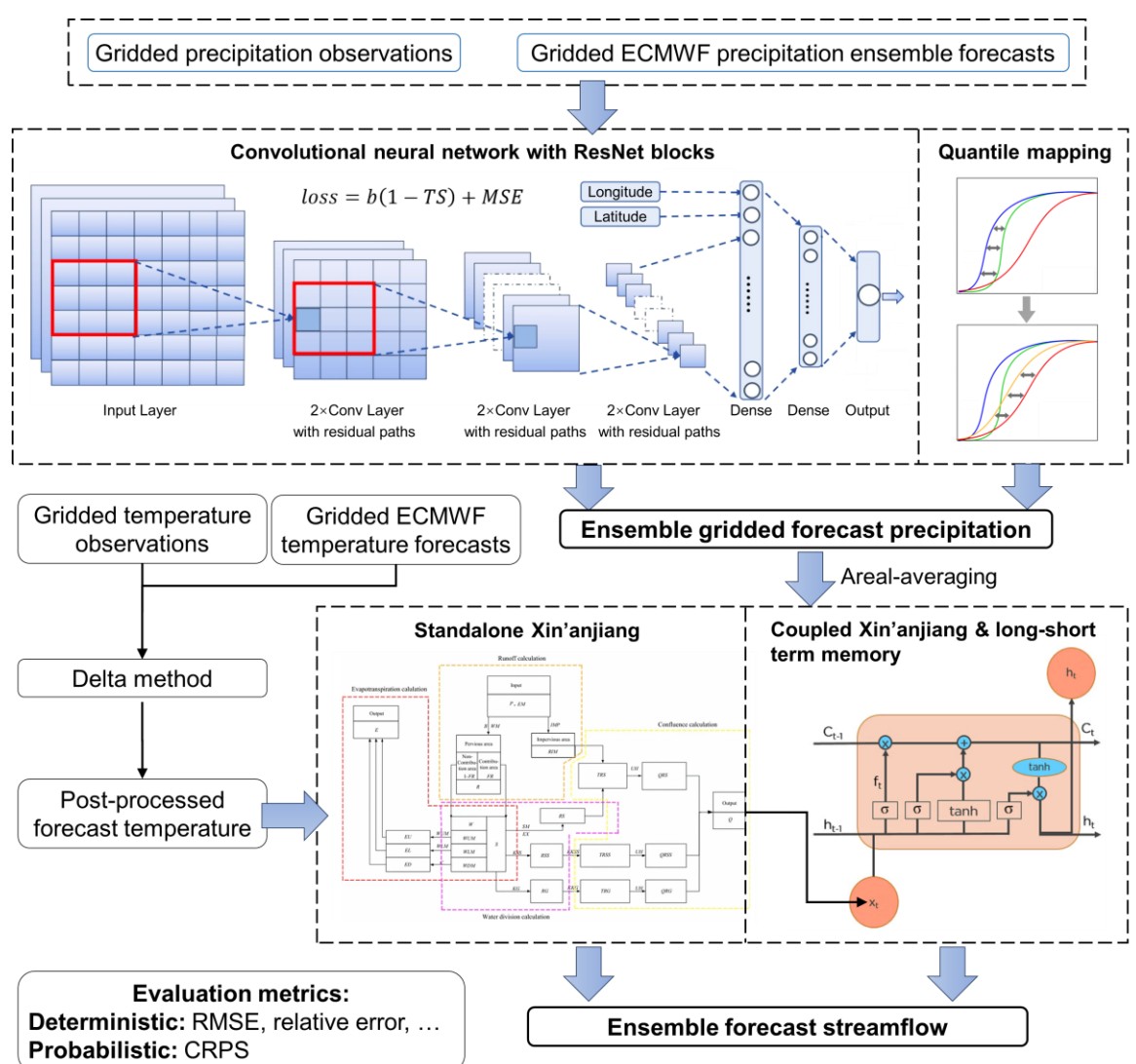

Figure 2. The workflow of this study.

### 3.2 Statistically downscaling of ensemble precipitation forecasts

### 3.2.1 Enhanced convolutional neural network

An ensemble of enhanced CNN models with ResNet blocks and a specialized loss function is established to learn the spatially
dependent relationship between fine-resolution local precipitation and coarse-resolution predictors from surrounding regions.
Specifically, it downscales a total of 10 ensemble members of the ECMWF S2S reforecasts from a 1.5° resolution to a 0.25°
resolution, using the CN05.1 reference precipitation dataset. The model takes spatially distributed inputs of 19 predictors,
including the surface elevation, convective precipitation, total precipitation at the surface level, and U/V wind components,
specific humidity, temperature, and geopotential height at 200/500/850 hPa pressure levels, from ECMWF forecasts. These



inputs cover a 3×3 area of coarse grid cells at 1.5° resolution, centered around the target fine grid cell at 0.25° resolution. Due to the square-shaped input structure of the CNN model, some ECMWF data from outside the basin boundary are included in the input. For the outputs, the predictand is the daily precipitation at a spatial resolution of 0.25°, and the CNN loops over each fine-resolution grid cell (0.25°) within the basin boundary, thereby generating a high-resolution precipitation forecast for the entire SR. Figure 2 presents the model structure, which primarily consists of convolutional layers, embedding layers, fully

connected layers, and residual paths. More details on the model structure are described as follows.

(1) Inputs to the network are predictors from a 3×3 grid patch (1.5° resolution) centered on the target grid cell (0.25° resolution). This patch includes a total of 19 meteorological variables, resulting in input arrays with sizes of 19×3×3. The spatial dimension of 3×3 is selected because it shows the best performance among four candidates of 1×1, 3×3, 5×5, and 7×7.

(2) The model includes three ResNet blocks, with each block containing two convolutional layers with 3×3 kernels and feature

maps of sizes 64, 32, and 16, respectively. Such block mitigates the vanishing gradient problem and improves computational efficiency for a moderately deep learning model as in our study by allowing the gradient to bypass certain layers. For each convolutional layer, the convolution procedure involves moving the kernels along the input spatial fields, with the dot product calculated between the inputs and the kernels to capture spatial features. The $l$ th feature map of the current convolutional layer $X_n^l$ is computed from the previous layer $X_{n-1}$ with $K$ feature maps through the convolutional operation as follows,

$$X_n^l = \text{ELU}\left( b_n^l + \sum_{k=1}^{K} W_n^{k,l} * X_{n-1}^k \right) \tag{1}$$

where $W_n^{k,l}$ are the convolutional kernels; $b_n^l$ is the bias for the $l$ th feature map; the symbol * denotes two-dimensional convolution. Here, we employ the exponential linear units (ELU) as the activation function, i.e.,

$$f(x) = \begin{cases} x & x > 0 \\ a(e^x - 1) & x \leqslant 0 \end{cases} \tag{2}$$

where $a$ is a hyperparameter to be estimated, $x$ is input to ELU function.

(3) To address spatial heterogeneity, embedding layers are introduced to convert the coordinate indices into latitude and longitude decimals (Rasp and Lerch, 2018). The outputs from these embedding layers are merged with the flattened outputs from the ResNet blocks, and this combined data is then fed into two fully connected layers before the output layer.

The Adam optimizer is used to train the CNN model with an early stopping technique to avoid overfitting. Specifically, to account for the small number of extreme precipitation samples, a specialized loss function that combines the Threat Score (TS)

and Mean Squared Error (MSE) is used in this study, i.e.,

$$loss = b(1 - TS) + MSE$$

$$MSE = \frac{1}{N}\sum_{i=1}^{N}(F_i - O_i)^2 \tag{3}$$

$$TS = \frac{H}{H + F + M}$$

where $b$ represents the weight of extreme precipitation in model training. $F_i$ is the $i$ th forecast data, $O_i$ is the $i$ th observation data, $N$ is the number of data. Meanwhile, $H$, $F$, and $M$ represent the hits, false alarms and misses, respectively. Note that the categorical indices used for calculating TS are discrete, which is not well-suited for training deep learning models. Thus, the differentiable formulations proposed by Larraondo et al (2020) and Lyu et al. (2023) are utilized in this study, i.e.,

$$H = (O > \alpha) \odot sigmoid(F - \alpha) \tag{4}$$

$$F = (O < \alpha) \odot sigmoid(F - \alpha) \tag{5}$$

$$M = (O > \alpha) \odot sigmoid(-F - \alpha) \tag{6}$$

$$sigmoid(x) = \frac{1}{1 + e^{-ax}} \tag{7}$$

in which $\odot$ means element-wise multiplication, the $(O > \alpha)$ and $(O < \alpha)$ are logical operations, which are 1 and 0 when the statement are true and false, respectively. $\alpha$ is the precipitation threshold that corresponds to the 90[th] percentile of observed precipitation of each grid cell for 2002-2019. The logical operations towards the $F$ (forecast) term are substituted with a sigmoid function, which represents a smooth transition between the Boolean values at the threshold point. In above expressions, $a$ and $b$ are hyperparameters that are determined following Lyu et al. (2023), see Table S1 in the supporting information for detailed values.

Another approach for improving extreme precipitation forecasts is to manually increase the number of heavy precipitation events within the training datasets. This approach is eventually not adopted in our study because it is found to degrade the sub-seasonal forecast accuracy of light precipitation events while not improving the accuracy of heavy precipitation events over the SR region (results not shown). A possible reason is that by doing so artificial disruptions are brought into the distribution of precipitation samples, which could possibly impair the generalization capability of models (You et al., 2023).

### 3.2.2 Quantile mapping

Quantile mapping is a widely used postprocessing technique and is able to effectively enhancing quantitative precipitation forecasts at the sub-seasonal timescale (Li et al., 2024). Therefore, the current study adopts QM as a benchmark to evaluate the proposed CNN-based model.

We implement QM using a non-parametric approach that adjusts the quantiles of the forecasted and observed data without assuming a specific distribution. Specifically, the empirical cumulative distribution functions (CDFs) of observed and forecasted daily precipitation are built respectively, and each percentile of the forecasted data is adjusted to match the corresponding percentile in the observed data. Dry days with a precipitation amount less than 0.1 mm are excluded from the derivation of CDFs (Gudmundsson et al., 2012). To match the 1.5° forecast resolution with the 0.25° reference dataset

resolution, the empirical CDFs are established separately for each 0.25° grid from the corresponding 1.5° forecast grid cell. Manzanas et al (2018), Cannon et al. (2015) and other studies have indicated the effectiveness of this implementation in improving the overall precipitation forecasts.

Here, the period from 2002-2015 is used to estimate the empirical CDFs, and these CDFs are then used to correct the EC forecasts in the test period of 2016-2019:


$$\tilde{p}_{QM} = O^{-1}[F(p_{EC})] \tag{8}$$

where $\tilde{p}_{QM}$ and $p_{EC}$ are the QM-based precipitation forecasts and the ECMWF raw precipitation forecasts, respectively. The QM is constructed separately for each lead time to account for forecast bias variations across different lead times. For each lead time, a single model is applied across all months, which is aligned with the structure of the CNN model built in this study.

### 3.3 Bias correction of temperature forecasts

In this study, we apply the widely used delta method to correct the ECMWF temperature forecasts for lead times of 1-30 days. We calculate the difference between observed and forecast temperature (i.e., the delta) for each lead time during May and August of 2002-2015 as a calibration period, and then apply a single delta model for each lead time to the forecast temperature during May and August of 2016-2019 as a validation period. Given that temperature is not the main focus of the paper, plus that temperature forecasts generally have less biases and much less hydrologic impacts than precipitation forecasts (as

discussed in Section 5.3), relevant evaluation results are provided in the Text S1 and Fig. S1 in the supporting information.

### 3.4 Hybrid hydrologic model of XAJ-LSTM

### 3.4.1 Xin'anjiang model

The XAJ model is a conceptual hydrological model (Zhao, 1992), which has been widely used to generate flood forecasts for humid and semi-humid regions of China. The lumped XAJ model consists of the evapotranspiration module, the runoff

generation module, the runoff partition module, and the runoff routing module (Hu et al. 2005). In this study, a modified version of XAJ model with snow accumulation and melting mechanisms is employed to simulate and forecast the daily streamflow of the SR at the sub-seasonal scale, which shows satisfactory accuracies for basins with snow melting runoff in our previous study (Tan et al., 2023).

### 3.4.2 Long-short term memory network

In this study, the LSTM model is employed as part of the modelling chain to simulate and predict the sub-seasonal streamflow. LSTMs have memory cells that are analogous to the states of a traditional dynamical system model (Kratzert et al, 2018), which make them practicable for simulating natural hydrologic systems. Compared with other types of RNNs, LSTMs perform better in coping with exploding and vanishing gradients, which enables them to learn the long-term dependencies between input and output arrays (Zhang et al., 2022). This is particularly desirable for modelling hydrological processes that have

relatively long-time dependencies as compared with input-driven processes such as direct surface runoff. For example, Kratzert et al. (2018; 2019) applied LSTMs to hydrologic modelling and show that the internal memory states of the network are highly correlated with observed snow and soil moisture states, even if no snow or soil moisture data were input to the models during training. This model feature allows accurate sub-seasonal hydrologic simulations in the SR where there is snow accumulation around the winter and spring.

**3.4.3 Model integration**

    The XAJ model employs daily precipitation and temperature of the two sub-basins to simulate the daily streamflow at the Shigu station. The model parameters are calibrated for the period of 1981-2015 and validated for the period of 2016-2019. The particle swarm optimization (PSO) approach is employed to optimize the parameters of the XAJ model, with the NSE as the objective function.

The LSTM model takes as input features a time sequence of daily precipitation $\mathbf{p_i} = \mathbf{p_i}[1], \mathbf{p_i}[2], ..., \mathbf{p_i}[N]$, daily temperature $\mathbf{t_i} = \mathbf{t_i}[1], \mathbf{t_i}[2], ..., \mathbf{t_i}[N]$ $(i = 1, 2)$ of two sub-basins, and XAJ-simulated daily streamflow $\mathbf{q} = \mathbf{q}[1], \mathbf{q}[2], ..., \mathbf{q}[N]$ over $N$ time steps. Each element of $\mathbf{p_i}, \mathbf{t_i}$ and $\mathbf{q_i}$, namely $\mathbf{p_i}[n], \mathbf{t_i}[n]$, and $\mathbf{q_i}[n]$, is a vector of input features for the past $n\_seq$ days and corresponds to the predictand $\mathbf{o}[n]$, the daily streamflow of Shigu station for timestep $n$. Here $n\_seq$ is an optimized hyperparameter representing the size of input features. In snow-affected regions such as the SR, it is typically set to a larger

value to account for the snow accumulation and melting processes, which can span hundreds of days. In our study, LSTM can also be considered a post-processing model of the XAJ model, similar to the CNN model as a post-processing model of the ECMWF forecast model. The LSTM model is trained by the Adam optimizer, and is 5-fold cross-validated using the Randomized Search approach for the period of 1981-2015 (see the Table S2 in the supporting information for details of model hyperparameters). The trained model is then tested for the period of 2016-2019.

**3.5 Evaluation Metrics**

    The precipitation forecasts are evaluated using the root-mean-squared-root (RMSE) and relative error (RE) for the ensemble mean and using the CRPS for the total 10 ensemble members. Specifically, we classify the 5-day daily precipitation less than and greater than the 90[th] percentile of all historic 5-day precipitation during 2002-2019 as light rain events and heavy rain events. The RMSEs are calculated for all rain events and heavy rain events to evaluate the forecasts in predicting common and

extreme events, which are both critical for sub-seasonal forecasts that need to inform agricultural planning and flood risk management.

The streamflow forecasts are evaluated using the RMSE, RE, and NSE for general trends, and the relative error of the maximum daily flow (REF) for extreme events.

## 4 Results

**4.1 Calibration and validation of the hybrid hydrologic model**

The conceptual XAJ model is calibrated for the period of 1981-2015 and validated for the period of 2016-2019. Results in Figure 3 indicate a satisfactory performance for the standalone XAJ model. The daily NSE values of simulated streamflow are 0.88 and 0.83 during the calibration and validation period. The relative error of streamflow is 1.0% and 2.6% during the calibration and validation period. The mean absolute error and relative error of simulated maximum daily flow are 844 $m^3$/s

and 17.0% during the calibration period, and are 379 $m^3$/s and 7.9% during the validation period, respectively.

The hybrid hydrologic model is calibrated for the period of 1981-2015 (corresponding to the calibration period of XAJ model and the training and cross-validation period of the LSTM model) and validated for the period of 2016-2019 (corresponding to the validation period of XAJ model and the testing period of the LSTM model). Fig. 3 depicts the simulated daily streamflow at Shigu during the validation period, as compared with observations. The results indicate that the daily NSE stands at 0.96

and 0.93 during the calibration and validation period, and the relative error stands at 1.7% and 2.8%, respectively. The mean absolute error and relative error of simulated maximum daily flow are 329 $m^3$/s and 7.5% during the validation period, respectively, indicating the model also has a satisfactory ability to simulate large flood events of the basin. By comparing the simulation accuracy of the standalone XAJ model with that of the hybrid model, it is found that the hybrid model can take advantage of the XAJ outputs and improve the streamflow simulations at Shigu station.

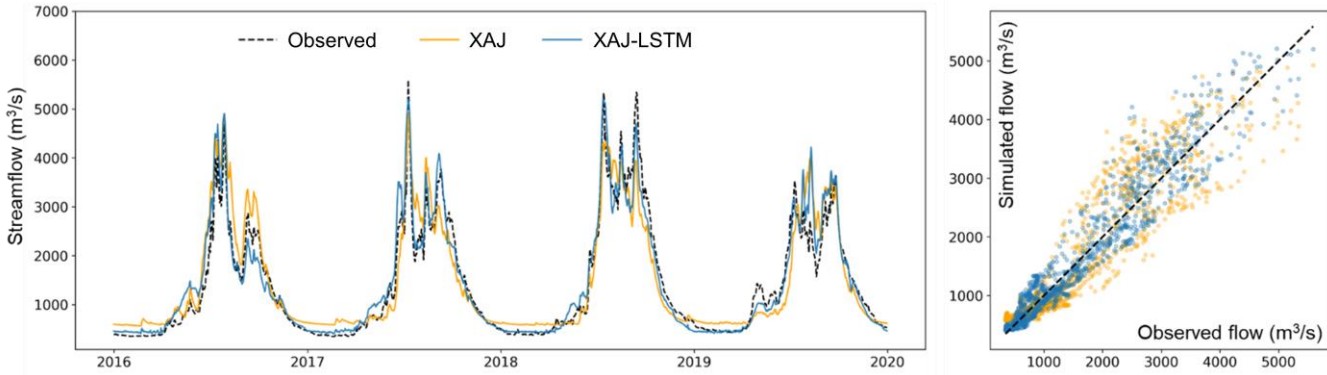


**Figure 3. Streamflow validation of the XAJ model and the XAJ-LSTM hydrologic model.**

## 4.2 Evaluation of sub-seasonal precipitation forecasts

The RMSE and RE of areal-averaged EC, EC-QM and EC-CNN precipitation at different lead time ranges for the period from 2016 to 2019 are provided in Figure 4, with the error bars representing the 25th-75th percentile interval.

Generally, the EC raw precipitation forecast skills decrease gradually with the increasing lead times and tend to be constant at a relatively low level for lead times of 15-30 days, which is also observed by Lyu et al. (2023) across Southeast China. The RMSE averaged over all lead times for the areal-averaged EC forecasts is 1.13 mm/day. The EC-QM effectively reduces RMSE at all lead times by an average of 0.12 mm/day (~11%), indicating the effectiveness of QM in improving precipitation at sub-seasonal scales. By contrast, the EC-CNN exhibits ~26% less RMSE compared to EC-QM for all lead times, which reduces the RMSE of EC forecasts by 0.38 mm/day (~34%). The RE shows a similar trend to RMSE, and the RE of EC, EC-QM, and EC-CNN are 27%, 36%, and 42% averaged over all lead times, respectively.

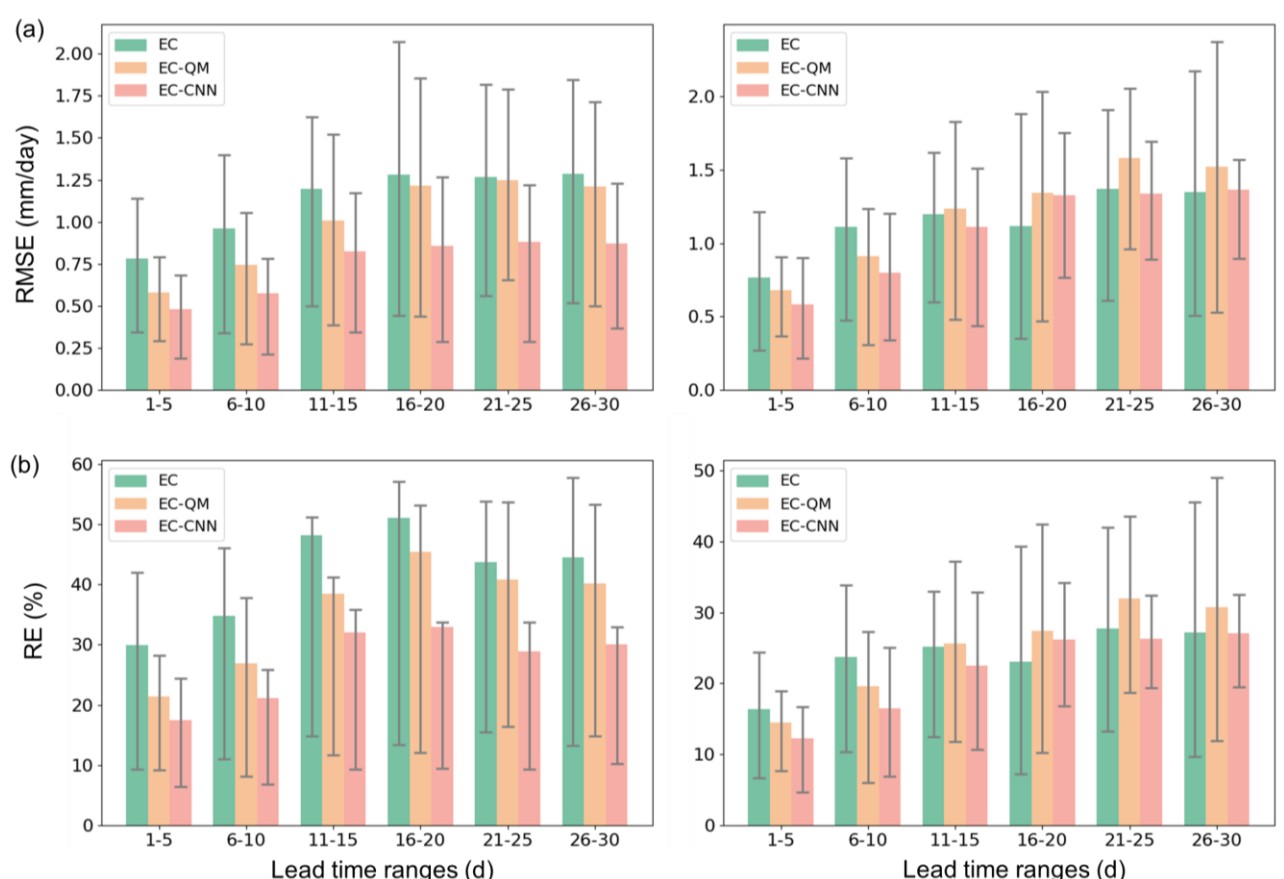

**Figure 4. (a) RMSE and (b) RE of all rain events (left) and heavy rain events (right) for the ensemble means of EC, EC-QM and EC-CNN 5-day precipitation forecasts at lead times of 1-5 days, 6-10 days, 11-15 days, 16-20 days, 21-25 days, 26-30 days. Error bar represents the 25th-75th percentile interval.**

In particular, the forecast accuracy is improved by EC-CNN forecasts to a relatively steady extent at all lead times, as the RMSE is reduced by 33%-34% for different lead time ranges. On the other hand, the RMSE improvements of EC-QM forecasts decrease rapidly with the increase in lead times. For example, the EC-QM reduces the RMSE of EC forecasts by 24% for the first 10 days, by 11% for the middle 10 days, and by 4% for the last 10 days. In addition, EC-CNN forecasts exhibit narrower 25th-75th percentile intervals of RMSE and RE across different initialized dates than raw EC forecasts and EC-QM forecasts, suggesting that the CNN model tends to produce precipitation forecasts with more stable skill metrics across different initialized dates. These results preliminarily demonstrate the superiority of proposed CNN method to the raw EC forecasts and the EC-QM forecasts.

Right panels in Figure 4 display the variations of RMSE for heavy rain events averaged over SR at lead times of 1-30 days, with the error bars representing the 25th-75th percentile interval. Generally, the RMSE of EC forecasts increases with the increasing lead times for both light and heavy rains. In terms of the heavy rain events, EC-QM shows no improvements as compared to the EC forecasts averaged over all lead times, with the RMSE increasing by ~5%, suggesting that QM has a limited ability to improve the forecast skills for extreme precipitation events in the SR. The EC-CNN generally shows slight improvements (~6%) in RMSE as compared to raw forecasts, and the RMSE of EC-CNN forecasts is smaller than that of EC-QM for all lead times, suggesting CNN exhibits advantages over QM for extreme events. This is particularly the case for lead times of 1-10 days, where the RMSE of EC-CNN (EC-QM) forecasts is 26% (14%) lower than that of EC forecasts. Overall, these results imply that, for heavy rainfall events, EC-CNN forecast has an advantage over the EC-QM forecast, and also shows a slightly better accuracy than the raw EC forecasts. The RE shows a similar trend to RMSE, as the RE of EC, EC-QM, and EC-CNN are 22%, 25%, and 24% averaged over all lead times. The EC-CNN effectively reduces the bias of heavy rain events at all lead times except for the 16-20 day range.

To investigate the spatial characteristics of precipitation forecasts, Figure 5 presents the spatial distribution on the RMSE of EC and EC-CNN forecast for lead times of 1-10, 11-20 and 21-30 days. It is clear that the EC-CNN improves the forecast skill of the raw ECMWF forecasts over the majority of the SR for all lead times. For example, the RMSE is reduced from 3-5 mm/day for EC forecasts to 1-2 mm/day for EC-CNN forecasts at the northern sub-basin for all lead times. Similar improvements can also be seen around the southern part of the basin, for example the southernmost part of SR sees RMSE over 10 mm/day for EC forecasts at lead times of 11-30 days but reduces to 6-7 mm/days for EC-CNN forecasts. In addition, by comparing Fig. R1b and Fig. R1c it can be seen that the EC-CNN shows larger improvements than EC-QM across the SR for all lead times. The above results indicate that EC-CNN not only improves the raw forecasts temporally, but also enhances their spatial accuracy across various regions of the SR. This basin-wise improvement allows for more reliable predictions across diverse hydrological zones within the SR, which could further benefit the hydrologic modelling.

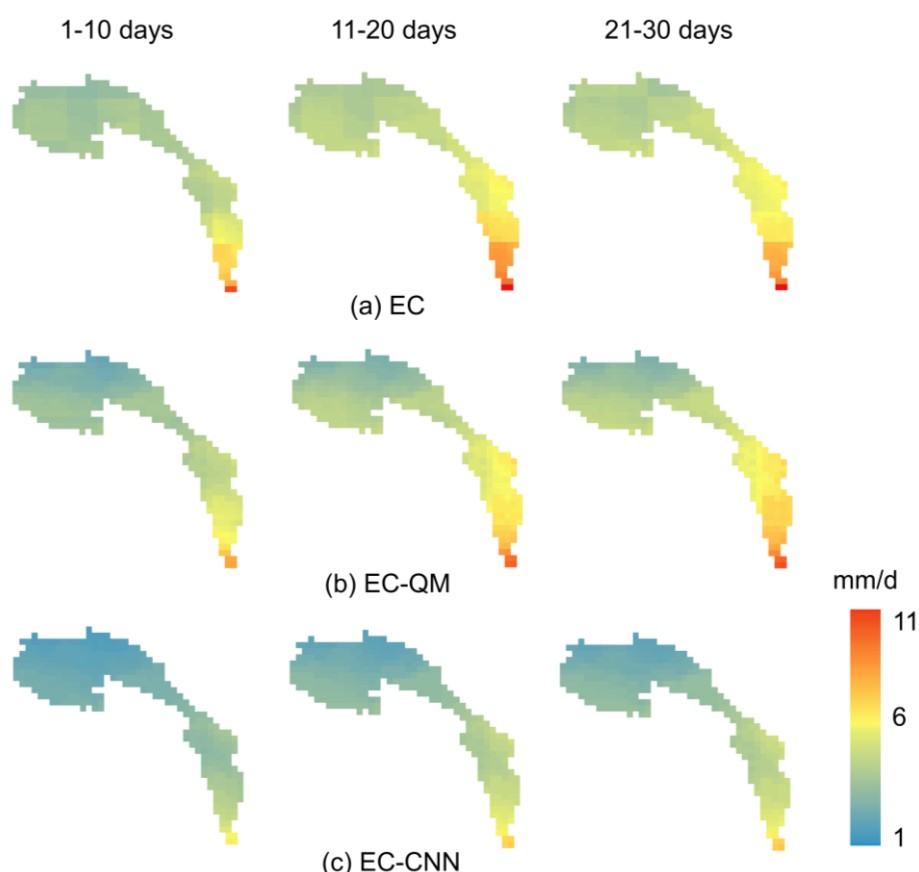

**Figure 5. The spatial distribution of RMSE for the ensemble means of (a) EC forecasts, (b) EC-QM, and (c) EC-CNN forecasts averaged over lead times of 1-10 days, 11-20 days, and 21-30 days during the test period.**

Figure 6 presents cumulative distribution functions (CDFs) of EC, EC-QM and EC-CNN forecasted precipitation averaged over the SR across all lead times. Notably, the EC-QM forecast consistently aligns well with the observed CDF across all lead times, which reflects the designed purpose of QM to match the empirical distribution through quantile mapping. This result demonstrates EC-QM is good at correcting the raw EC forecast to follow the observed distribution closely, despite an overall large RMSE and RE compared to EC-CNN.

The EC-CNN forecast shows improvement over the EC forecasts and EC-QM forecasts by better approximating the observed CDF for the first 15 days. However, as compared to EC and EC-QM, the EC-CNN begins to deviate more significantly from the observed CDF as lead times increase. Specifically, the EC-CNN forecast appears to concentrate around medium precipitation values for 16-30 days, which underestimates the frequency of both lighter and heavier precipitation events. This pattern suggests that, while EC-CNN improves the overall accuracy of light and heavy rains of the raw EC forecasts at all lead times (Figure 4), the distributional accuracy, particularly for extreme precipitation events, may be compromised over extended lead times. A discussion on the possible cause of biases in CDF for different forecasts is provided in Section 5.4.

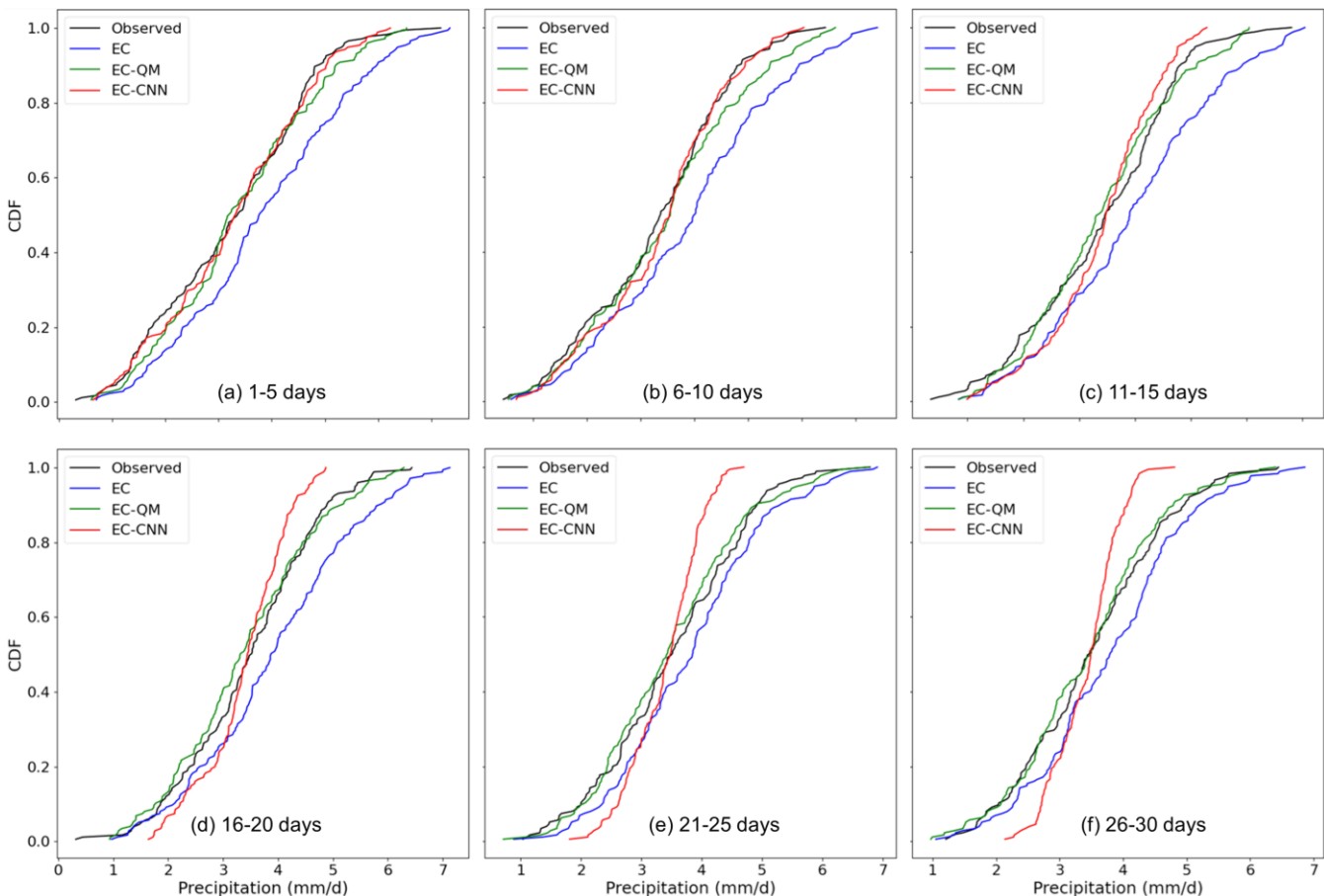

**Figure 6.** The cumulative distribution function (CDF) of areal-averaged precipitation for observed precipitation, the ensemble means of EC, EC-QM, and EC-CNN forecasts at different lead times during the test period.

Figure 7 presents the CRPS of EC, EC-QM and EC-CNN forecasted precipitation averaged over the SR across the lead times. CRPS evaluates how close the ensemble forecast distribution is to the observed value, and a value close to zero means a better ensemble forecast. As can be seen in the figure, the CRPS for EC is around 1.1 mm/day at lead times of 1-10 days and increases to around 1.4 mm/day at lead times of 11-30 days. The EC-QM reduces the CRPS by an average of around 0.1 mm/day at all lead times, indicating an improvement in the probabilistic calibration and sharpness of the ensemble forecasts. The EC-CNN further reduces the CRPS for most of the lead times as compared to the EC-QM, especially for the first 5 days where the CRPS is 0.4 mm/day lower than EC forecasts and 0.2 mm/day lower than EC-QM forecasts. This shows that the EC-CNN has an enhanced capability of representing the range of possible outcomes and improving the overall reliability in probabilistic forecasting. Such an advantage also offers better decision-making insights under uncertainty, which is favorable for risk management and planning across various time horizons.

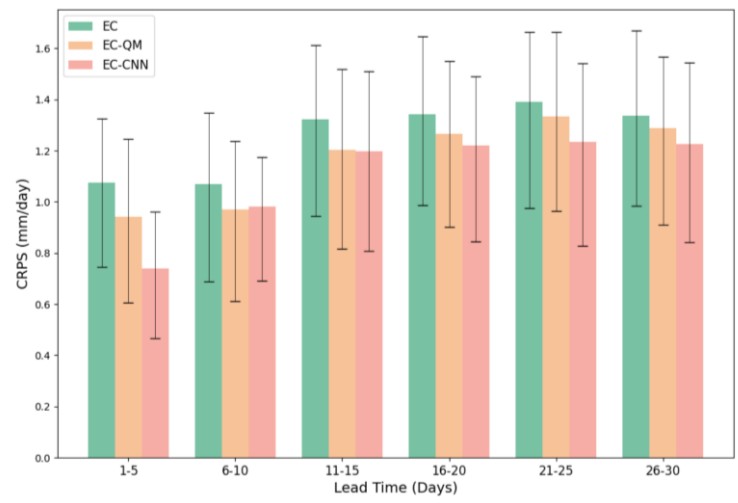

**Figure 7. The CRPS of areal-averaged precipitation for the EC, EC-QM, and EC-CNN ensemble forecasts at different lead times during the test period. Error bar represents the 25th-75th percentile interval.**

### 4.3 Evaluation of sub-seasonal streamflow forecasts

Figure 8 shows the RE, REF, RMSE and NSE of the XAJ-LSTM and XAJ streamflow forecasts driven by EC, EC-QM, and EC-CNN precipitation forecasts, respectively. Results indicate that, for the XAJ-LSTM hybrid model, the accuracy metrics of streamflow forecasts decrease as the lead times increase for all precipitation forecasts. For example, the RE (REF) of streamflow forecasts driven by EC precipitation increases from around 11.6% (14%) for 1-10 days to 24% (23%) and 28% (27%) for 11-20 days and 21-30 days, respectively. The EC-QM (EC-CNN) forecasts reduce the RE of the EC forecasts by approximately 6.9% (16.4%), 13.8% (32.5%), and 12.5% (26.8%), respectively, and reduce the REF of the EC forecasts by approximately 14.3% (28.6%), 10.4% (35.7%), and 13.3% (25.6%), respectively. It is noted that the improvements brought by EC-QM and EC-CNN are larger in lead times of 11-20 days than in 1-10 days, which was also observed by Zhang et al. (2023), Lyu et al. (2023), and Li et al. (2024) in specific cases.

For the standalone XAJ model, the RE (REF) of streamflow forecasts driven by EC raw forecasts increases from around 18% (21%) for 1-10 days to 31% (33%) and 34% (31%) for 11-20 days and 21-30 days, respectively. The EC-QM (EC-CNN) forecasts reduce the RE of the EC forecasts by approximately 5.6% (9.4%), 8.1% (18.4%), and 0.6% (12.4%) for the 1-10, 11-20, and 21-30 day lead times, respectively. For the relative error of maximum daily flow, the EC-QM (EC-CNN) forecasts reduce it by approximately 9.4% (26.7%), 11.6% (24.8%), and 0.9% (5.8%) for each corresponding lead time range.

The above results indicate that (1) the streamflow and flood biases are smaller for XAJ-LSTM than for XAJ for all lead times, and (2) improvements in EC-CNN and EC-QM precipitation forecast enhance the streamflow forecast accuracy more effectively for the XAJ-LSTM model than for the XAJ model. Notably, the EC-CNN (EC-QM) driven XAJ-LSTM streamflow forecast sees 20%‑31% (6%‑12%) less RMSE than that driven by EC forecasts over different lead time periods. On the other

hand, the EC-CNN (EC-QM) driven XAJ streamflow forecasts only see 7%-11% (1%-7%) less RMSE compared to that driven by EC, reflecting a much less improvement compared to those driven by XAJ-LSTM. A similar trend can also be observed for NSE, as XAJ-LSTM shows more improvement than XAJ when EC precipitation forecasts are replaced by EC-CNN and EC-QM precipitation forecasts. This result suggests that improving sub-seasonal precipitation forecasts may not necessarily translate to a streamflow improvement, because it is related to not only the skill of precipitation forecasts but is also to a large extent the hydrologic model.

It is also noted that, despite the NSE values improve with the EC-CNN precipitation forecasts, they are mostly negative for both hydrologic models. This suggests further improvements may be required to achieve a more accurate hydrologic forecast at sub-seasonal scales. A discussion on this aspect is provided in Section 5.4.

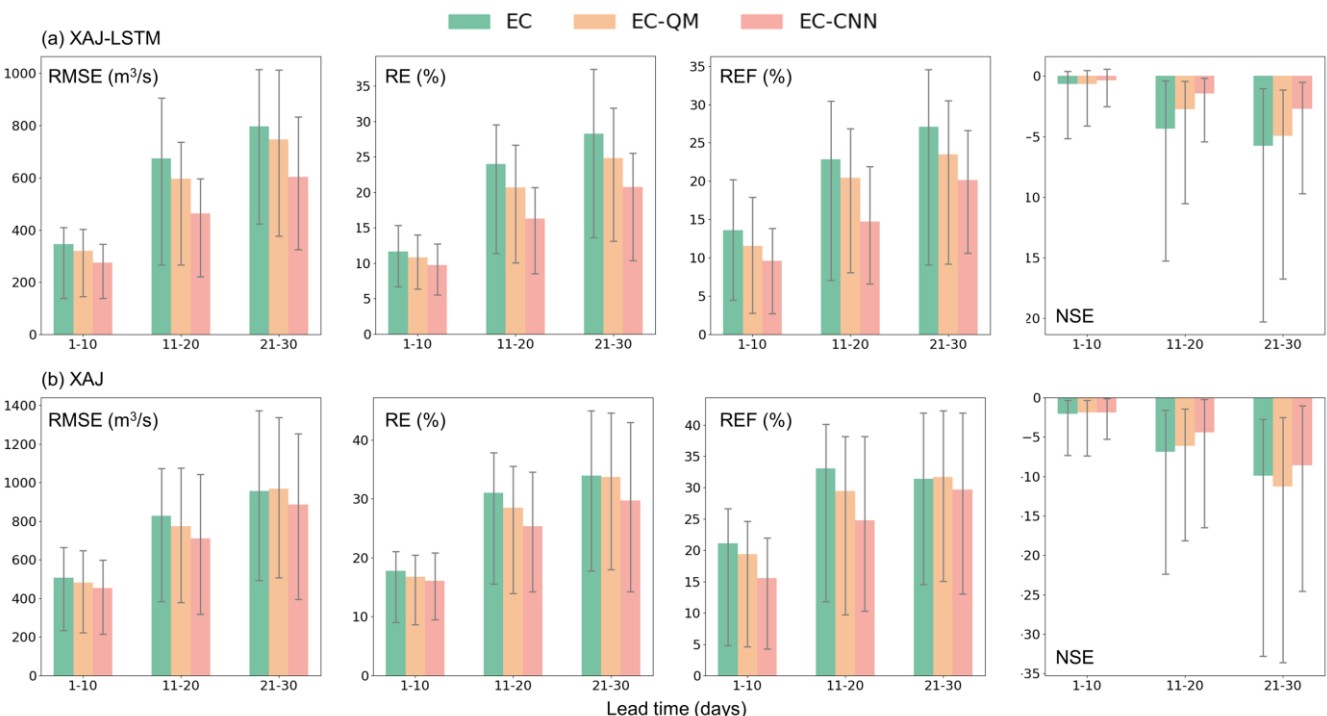

**Figure 8. The RMSE, RE, REF, and NSE for the (a) XAJ-LSTM and (b) XAJ streamflow forecasts driven by the ensemble means of EC, EC-QM, and EC-CNN forecasts for lead times of 1-10 days, 11-20 days, and 21-30 days. Error bar represents the 25th-75th percentile interval.**

Figure 9 presents examples of XAJ-LSTM streamflow ensemble forecasts initialized on different dates, with the RE of total forecast flow presented in each subplot. In most cases, the EC-CNN forecasts can reduce the streamflow bias in a more flexible manner than the EC-QM forecasts, resulting in more accurate overall streamflow predictions across different dates. For example, the CNN model decreases (increases) the EC precipitation and hence the forecast streamflow issued on June 06, 2017 (August 08, 2019), which improves the forecast skills in both cases. The EC-CNN reduces the relative error from 22.1% of

raw EC forecasts to -2.5% for the 30-day streamflow forecast issued on June 06, 2017, and reduces the relative error from -

21.5% of raw EC forecasts to -9.2% for the 30-day streamflow forecast issued on August 08, 2019. On the other hand, the QM

reduces the precipitation and hence the streamflow forecasts for both dates, which reduces the relative error to 11.0% on June

2017 but increases the relative error to -27.3% on August 2019. Similarly, for the forecast issued on Aug 11, 2017, EC-CNN

decreases the EC precipitation for lead times of 1-20 days and increases it for lead times of 21-30 days, which alleviates the

streamflow overestimation on late August and underestimation on early September. However, EC-QM consistently predicts

lower precipitation and hence streamflow compared to EC forecasts at all lead times, worsening the underestimation in early

September.

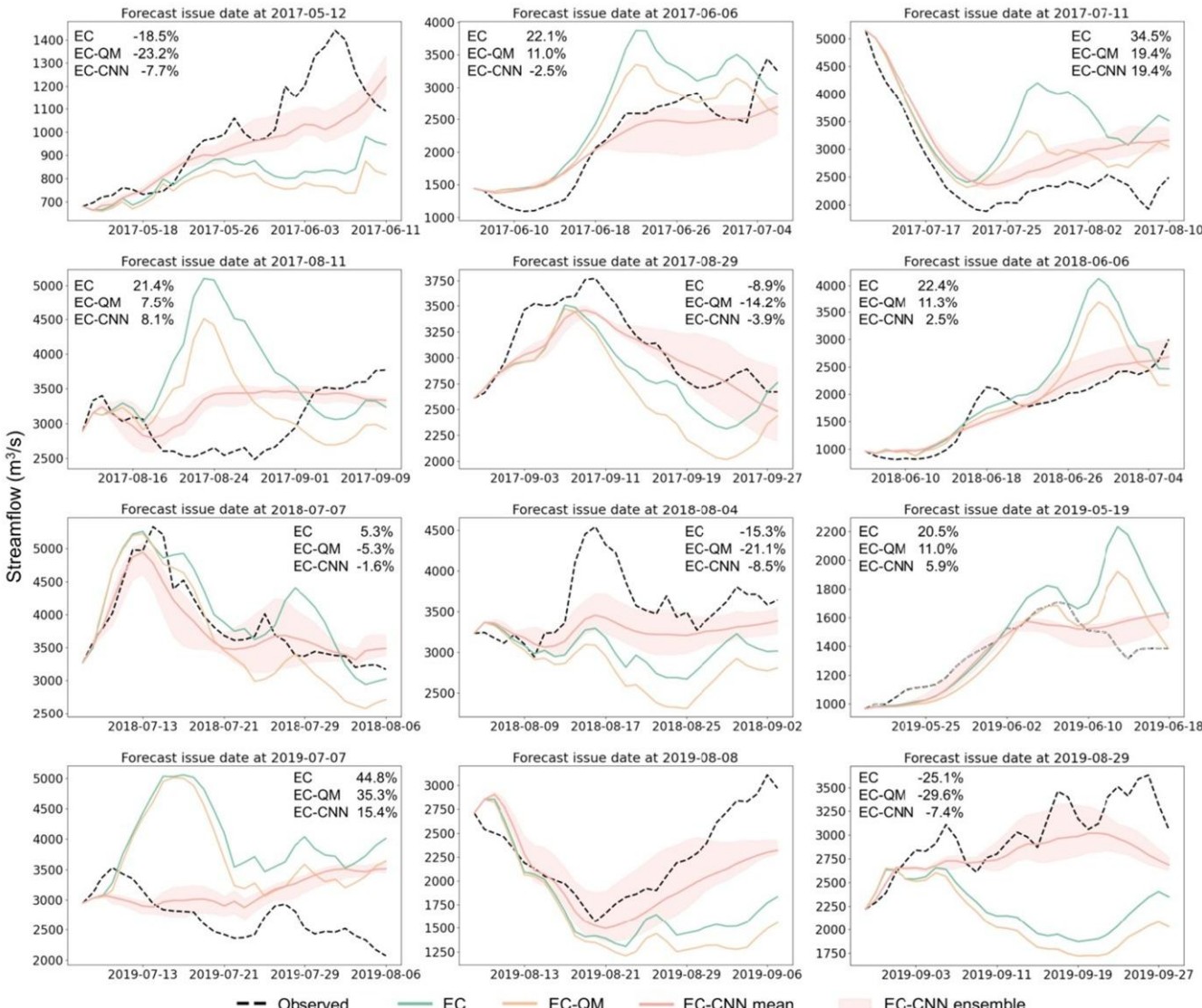


**Figure 9. Examples of sub-seasonal XAJ-LSTM streamflow forecasts for a lead time of 30 days driven by EC, EC-QM, and ensemble EC-CNN precipitation forecasts. Shaded areas represent the 25th-75th percentile. The relative error of total forecast flow is shown in each subplot.**

Figure 10 presents the CRPS of EC, EC-QM and EC-CNN driven XAJ-LSTM streamflow forecasts at Shigu across the lead times. The CRPS for EC is around 120 m³/s at lead times of 1-5 days and increases rapidly to around 300 m³/s at lead times of 6-10 days, and further to 500 m³/s at lead times of 21-30 days. The EC-QM reduces the CRPS by an average of around 65 m³/s at all lead times, and the EC-CNN further reduces the CRPS for most of the lead times as compared to the EC-QM, especially

for the lead times of 6-20 days where the CRPS is about 60 m³/s lower than EC forecasts and 25 m³/s lower than EC-QM forecasts. However, for lead times of 26-30 days, the CRPS of EC-CNN is slightly larger than that EC-QM, indicating the advantage of EC-CNN in ensemble forecasting is not evident for extended forecast lead times. Nevertheless, the EC-CNN improves the overall reliability in probabilistic streamflow forecasting for all lead times as compared to EC and for most lead times as compared to EC-QM, which can benefit the downstream water resources management under uncertainty.

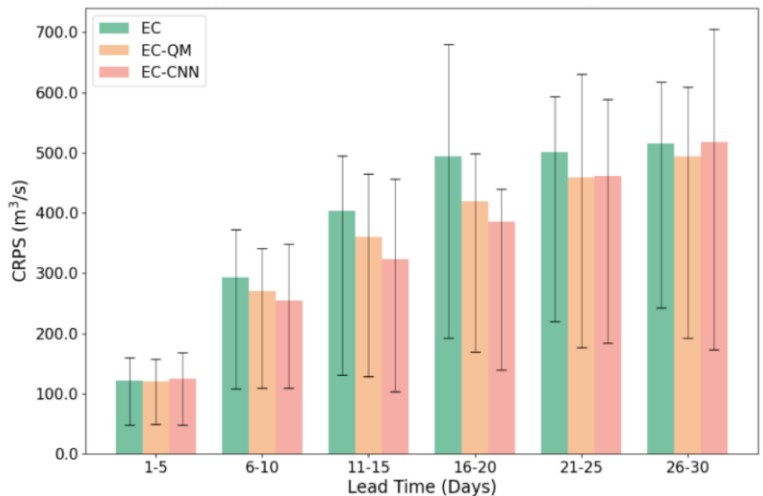


**Figure 10. The CRPS of streamflow forecasts driven by the EC, EC-QM, and EC-CNN ensemble forecasts at different lead times during the test period. Error bar represents the 25th-75th percentile interval.**

## 5 Discussion

### 5.1 Deep learning models can outperform traditional statistical downscaling methods in both mean and extremes

Traditional post-processing methods for precipitation forecasts often rely on local precipitation forecasts as the sole predictor, which can limit their ability to fully utilize the spatial information embedded in raw forecasts (Sun et al., 2021). In this study, an ensemble of enhanced CNN post-processing models with ResNet blocks and a weighted loss function specialized on extreme events is established to investigate its potential to overcome these limitations by establishing multi-dimensional relationships between atmospheric circulation predictors and local precipitation.

We compare the CNN model with the commonly used quantile mapping (QM) bias correction method. It is noted that, for several lead time ranges, EC-QM forecasts show no improvements in RMSEs compared to those of raw EC forecasts. This was also observed by some of the recent studies using QM for statistical downscaling precipitation. For example, Li et al. (2023), Huang et al. (2022), and Mao et al. (2015) show that, while QM is generally effective in adjusting model bias towards observations, it does not always lead to improvements of the forecast accuracy. One plausible reason could be the limited

applicability of the relatively simple QM method. Specifically, QM primarily adjusts the distribution of forecasted values towards the distribution of historical observations, with no account of the atmospheric conditions associated with those forecasts. However, physics-based numerical weather predictions involve complex and nonlinear errors that QM may not be able to fully correct.

On the other hand, the CNN model improves the RMSE and RE of forecast precipitation at all lead times, and outperforms

QM in terms of capturing the general trends, predicting extreme precipitation events, and approximating the probabilistic distribution at sub-seasonal scales. This superior performance is likely due to the specialized loss function that balances the prediction of light rain events and extreme events by incorporating the mean squared error and the threat score. This balance is crucial for sub-seasonal forecasts, where both event types impact water resource management. In general, the CNN structure used in our study is not only effective and easy to implement but also more computationally efficient than more complex CNN

variations like SmaAt-UNet. Nevertheless, newer variants may better leverage multi-scale spatial information and incorporate multiple auxiliary predictors relevant to local weather conditions (Rasp and Lerch, 2018; Peng et al., 2020; Baño-Medina et al., 2020). Future research will focus on integrating these variants with new loss functions to achieve more desirable forecast outcomes.

The rapid development of AI-based weather prediction models in recent years, such as Pangu and GraphCast, has also

demonstrated the potential of these models to achieve forecast skills comparable to state-of-the-art physics-based models (Bi et al., 2023; Lam et al., 2023). In comparison, our CNN-based statistically downscaled model of ECMWF precipitation forecasts offers improved sub-seasonal forecast skills with significantly lower computational resources, which could make it a practical and efficient tool for operational use in local meteorological or water agencies to provide high-quality forecasts and issue early warmings. Our results also underscore the potential of combining advanced AI techniques with physics-based

forecasting methods to achieve superior performance and operational efficiency in weather prediction.

## 5.2 Better sub-seasonal precipitation forecasts may not guarantee better streamflow forecasts

The evaluation of sub-seasonal streamflow forecasts in Section 4.3 reveals a complex relationship between precipitation forecast accuracy and streamflow forecast performance. For example, the results presented in Figure 8 demonstrate that, while improvements in precipitation forecasts generally lead to better streamflow forecasts, this relationship is not straightforward

and can be influenced significantly by the choice of hydrologic model.

For example, a notable finding is that the hybrid XAJ-LSTM model shows much more substantial streamflow improvements with better precipitation forecasts compared to the standalone XAJ model. Specifically, the XAJ-LSTM model, which combines the strengths of LSTM networks and the XAJ hydrologic model, benefits significantly from the enhanced accuracy of EC-CNN forecasts. This model demonstrates a considerable reduction in RMSE for streamflow predictions over various

lead times. On the other hand, the standalone XAJ model exhibits marginal improvements when driven by the same enhanced precipitation forecasts.

This disparity suggests that while advanced precipitation forecasts provide more accurate inputs, the ability of hydrologic models to effectively utilize these inputs is crucial. Similar findings are also reported by Valdez et al. (2022) for lead times of 7 days, who attribute the potential degradation of streamflow forecasts to other sources of uncertainties that may cancel out

the added values of precipitation forecast improvements. The integration of machine learning (LSTM) and physical process representations (XAJ) allows it to better capture the long-term dependencies in hydrological processes, making it more responsive to the quality of precipitation forecasts at sub-seasonal scales. This synergy between the two models enables to leverage the strengths of both conceptual understanding and data-driven prediction, which can also be extended to other basins with similar hydrological characteristics for addressing the sub-seasonal forecasting challenges.

**5.3 Attribution of the XAJ-LSTM streamflow forecast error**

To identify the possible sources of error for the XAJ-LSTM streamflow forecasts, an error decomposition method is employed to break down the total forecast error into its constituent parts. The specific contributions of each error source are isolated by calculating the RMSE of the ensemble-mean streamflow forecast driven by observed precipitation and temperature (i.e., the hydrologic modelling error, $E_m$), the RMSE of the ensemble-mean streamflow forecasts driven by forecast precipitation and

observed temperature (i.e., the precipitation forecast error, $E_p$), and the RMSE of the ensemble-mean streamflow forecasts driven by observed precipitation and forecast temperature (i.e., the temperature forecast error, $E_t$). Note that the total error between the observed streamflow and the forecast streamflow driven by forecast precipitation and temperature may not equal to the sum of $E_m$, $E_p$, and $E_t$, due to the interacting effects between multiple sources of error. This is manifested by the analysis result that the individual contributions of $E_p$, $E_m$, and $E_t$ to the total error add up to a value greater than 100%, indicating that

there is a compensatory effect between multiple sources of error that reduces the total error.

Figure 11 depicts the individual contribution of $E_p$, $E_m$, and $E_t$ to their combined error. In general, the hydrologic modelling error $E_m$ dominates for lead times of 1-3 days, accounting for over 50% of the three sources of error combined. The ratio of $E_m$ decreases rapidly with the increase in lead times, and reaches a steady value of around 0.3 after the lead time of 15 days. The contribution ratio of precipitation forecasts error $E_p$ rise rapidly for lead times of 1-7 days, and stands at a steady value of

around 0.6 after the lead time of 15 days. The temperature forecast error $E_t$, while present, has a less pronounced impact

compared to $E_m$ and $E_p$, accounting for 5%-10% of the combined error. This is an expected result as precipitation generally impacts the streamflow more significantly than temperature.

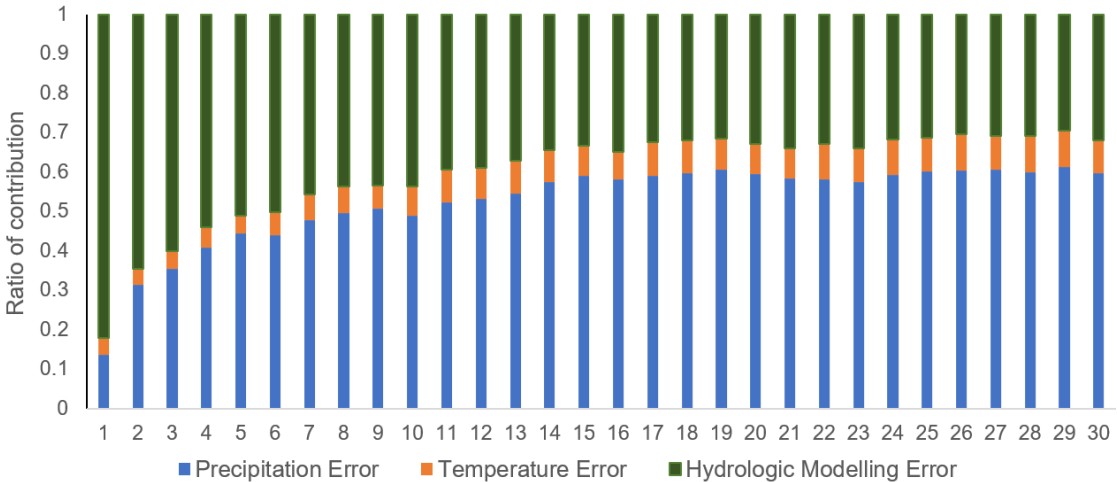

**Figure 11. Contribution of precipitation forecast errors, temperature forecast errors, and hydrologic modelling errors to their combined error.**

### 5.4 Limitations of this study

One limitation of the proposed EC-CNN model is its reduced accuracy in capturing the CDF over lead times extending beyond 15 days. While EC-CNN outperforms in terms of RMSE and RE for both light and heavy rainfall events (Figure 4), its CDF deviates from observed patterns for these longer lead times. In contrast, EC and EC-QM tend to align more closely with observed CDFs at these lead times. We attribute this to the following possible reason.

Over extended lead times (>15 days), extreme precipitation values of EC and EC-QM forecasts tend to be more incorrectly assigned to specific times, leading to, for instance, storm-level precipitation predictions on dry days or nearly zero precipitation predictions on storm days. While these misplaced extremes inflate daily errors, they still allow the overall CDF of the forecast to retain both high and low ends of the precipitation distribution. The EC-CNN model, however, is designed to minimize daily errors by adjusting outliers and bringing exaggerated values closer to moderate levels, thereby reducing large forecast errors. For example, when EC forecasts a high precipitation event on a dry day, EC-CNN mitigates this by lowering the extreme to a more typical value. While this adjustment helps decrease RMSE and RE, it compresses the distribution toward the center by reducing the frequency of both extreme high and low precipitation values. This approach limits the model ability to capture the full distribution, resulting in a CDF that is overly concentrated around moderate values.

Another limitation is the relatively low Nash-Sutcliffe Efficiency (NSE) values in streamflow forecasts, especially over extended lead times. Despite improvements in streamflow forecast accuracy due to EC-CNN and the XAJ-LSTM, NSE values are found predominantly negative. This can be primarily due to the inaccuracies in the precipitation forecasts that drive these

hydrologic models, as high NSE values require precipitation inputs to be accurate both spatially and temporally. Improving the hydrologic model alone is unlikely to address this issue substantially, as it depends heavily on input accuracy (Section 5.3).

Nevertheless, with our proposed coupled EC-CNN and XAJ-LSTM framework, the overall relative error of forecast flow can be reduced to ~10% for the next 10 days and ~20% for the next 30 days (Figure 7), which can provide implications for water management and disaster prevention. Future improvements in NSE could focus on refining precipitation forecasting with more advanced AI models. Additionally, LSTM model training on multiple basins and fine-tuning on specific target basins could further enhance the streamflow accuracy over extended lead times (Kratzert et al., 2019).

**6 Summary and conclusions**

This study proposes a deep learning based modelling framework for sub-seasonal hydrometeorological forecasts (i.e., precipitation and streamflow) for a lead time of up to 30 days. The framework couples (1) an ensemble of enhanced CNN models with ResNet blocks for statistically downscaling ECMWF raw precipitation ensemble forecasts to (2) a hybrid hydrologic model integrating the conceptual XAJ model and LSTM for streamflow forecasting. The CNN models incorporate

a specialized loss function that combines the continuous form of TS and MAE.

By applying the modelling framework to the source region of the Yangtze River Basin, we show that the CNN-based downscaling model exhibits advantages over quantile mapping in improving the precipitation forecasts in terms of the general trends, extreme events, and ensemble distribution. The CNN-based model consistently outperforms the raw ECMWF forecasts and the traditional QM approach across all lead times, achieving an average RMSE value around 30% lower than both forecasts.

This improvement is also noted in extreme precipitation events, as demonstrated by approximately 6% and 10% lower RMSE of the CNN for heavy rain events as compared to raw forecasts and QM forecasts.

With these precipitation forecasts serving as meteorological drivers of a hybrid XAJ-LSTM hydrologic model, it is found that CNN-based models can reduce the relative error of streamflow forecasts by 16%-33% compared to raw precipitation forecasts, particularly for longer lead times. This outperforms QM, which reduces the relative error of streamflow by 7%-14% compared

to raw precipitation forecasts. The CNN-based precipitation forecasts also prove effective in deriving more reliable streamflow forecasts during extreme hydrological events (such as floods) for the XAJ-LSTM model, with the average relative error of maximum daily flow reduced by 26%-36%. However, for the standalone XAJ model, the streamflow forecasts show marginal improvements with the same CNN enhanced precipitation forecasts. This highlights the importance of understanding the effectiveness of the hydrologic model as part of the sub-seasonal hydrometeorological modeling chain.

From a practical perspective, the proposed modelling framework is computationally efficient, requiring lower computational resources compared to fully AI models, traditional dynamic downscaling methods, and distributed hydrologic models. This makes it a viable tool for operational use in local meteorological and water management agencies to provide more accurate

forecasts and issue early warnings. This study also shows the potential of combining advanced AI techniques with traditional hydrologic modelling approaches to achieve superior performance in sub-seasonal hydrometeorological forecasting, offering a robust and adaptable solution for effective water resources management and disaster preparedness.

**Code Availability Statement**

The CNN model for statistically downscaling and bias correcting the ECMWF raw forecasts is deposited at Zenodo repository (DOI: 10.5281/zenodo.12664798). The LSTM and the hybrid hydrologic model are developed and configured using the NeuralHydrology package (Kratzert et al., 2022), available at https://neuralhydrology.readthedocs.io/en/latest/index.html.

**Data Availability Statement**

The ECMWF forecast data and observed precipitation and temperature data are all deposited at Zenodo repository (DOI: 10.5281/zenodo.12664851).

**Author Contributions**

ND contributed to the research design, compiled the dataset, conducted the data processing, analysis, and manuscript preparation. HH contributed to the research design, data processing, and code development. MY, JW, SX, and HK contributed to the manuscript editing.

**Conflict of Interest**

The authors declare that they have no conflict of interest.

**Acknowledgements**

Ningpeng Dong received funding for this study through the National Key Research and Development Program of China (2023YFC3081000) and the National Natural Science Foundation of China (42401053). Jianhui Wei is financially supported by the German Federal Ministry of Science of Education (BMBF) through funding of the KARE_II project (01LR2006D1). This study is also funded by China Power Construction Corporation Technology Project (DJ-HXGG- 2021-04) and Key R&D Plan Project in Yunnan Province (202203AA080010).

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
