# Peer review of "Deep learning based sub-seasonal precipitation and streamflow ensemble forecasting over the source region of the Yangtze River"

_Hydrology and Earth System Sciences, 2024_

## Author Comment (AC1)

**Response to Reviewer #1**

Thank you for your comments. Below are our replies to each of your comment.

We believe that the planned changes will improve the clarity and significance of our manuscript.

**Major Comments:**

**Comment:** In introduction, L45-70 introduce the advantages and disadvantages of dynamic downscaling and statistical downscaling, which are well known to us, and just to illustrate a few deep learning methods. Failure to highlight the research focus of this paper, i.e., the progress of research on traditional statistical downscaling and statistical downscaling combined with deep learning.

**Reply:** We appreciate the reviewer's comment. We will revise the introduction to emphasize the progress made in statistical downscaling methods, particularly when combined with deep learning techniques. This will better align the introduction with the specific research focus of our paper.

**Comment:** I do not understand the role of L70-75, it seems to describe weather prediction models like Pangu and GraphCast which can achieve entirely through deep learning have demonstrated the potential to achieve forecast skills comparable to state-of-the-art numerical weather prediction systems. Is this relevant to the research in this paper?

**Reply:** The mention of Pangu and GraphCast was intended to introduce the state-of-the-art deep learning models for weather prediction in a broader context. To bridge the gap in deep learning models from the broader context to the focus of our paper, we will revise these lines to ensure that this literature review remains directly relevant to our study.

**Comment:** About result, some of the conclusions seem too brief. L255-260 have no results about the relative error stands, the mean absolute error and relative error of simulated maximum daily flow.

**Reply:** We acknowledge the need for more detailed descriptions of results. We will expand this section to include the results of relative error and mean absolute error of the simulated maximum daily flow against the observations.

**Comment:** The author writes "while the RMSE of EC-QM forecasts sees a relatively steady reduction over all lead times"(L280-285), but from the Fig.4, we can clearly find that EC-QM showed more RMSE compared to EC for the lead times of 23-24 days, 27 days. The author can explain the reason? And please draw the spatial distribution of the bias of EC-CNN, EC-QM and EC because it mentioned in Line 310.

**Reply:** We agree with the Reviewer that the QM method does not reduce the RMSE in an exactly steady manner with lead time. In particular, for lead times such as 23, 24 and 27 days, QM-based forecasts show a slightly larger RMSE than that of raw forecasts. Our finding is in line with these of recent studies using QM for statistical downscaling precipitation. For example, Li et al. (2023), Huang et al. (2022), and Mao et al. (2015) show that, while QM is generally effective in adjusting model bias towards observations, it does not always lead to improvements of the forecast accuracy.

One plausible reason could be the limited applicability of the relatively simple QM method. Specifically, QM primarily adjusts the distribution of forecasted values towards the distribution of historical observations, with no account of the atmospheric conditions associated with those

forecasts. However, physics-based numerical weather predictions involve complex and nonlinear errors that QM may not be able to fully correct. For example, for lead time of 23 days, the QM consistently predicts lower precipitation than raw forecasts for light rain events across all initial dates, while the actual biases are positive for some initial dates and negative for other dates. Another reason could be that QM does not explicitly account for errors in the spatial dependencies of precipitation field, which can also contribute to the increased RMSE in specific lead times.

We will revise the manuscript to include the above-mentioned explanations and to add relevant references accordingly. We will provide comprehensive discussions about the limitations of QM in the context of sub-seasonal forecasting. For visualization, a spatial plot of forecasting error will be added.

**References**

Li, X., Wu, H., Nanding, N., Chen, S., Hu, Y., & Li, L. (2023). Statistical Bias Correction of Precipitation Forecasts Based on Quantile Mapping on the Sub-Seasonal to Seasonal Scale. *Remote Sensing, 15*(7), 1743. https://doi.org/10.3390/rs15071743

Huang, Z., Zhao, T., Xu, W., Cai, H., Wang, J., Zhang, Y., Liu, Z., Tian, Y., Yan, D., & Chen, X. (2022). A Seven-Parameter Bernoulli-Gamma-Gaussian Model to Calibrate Subseasonal to Seasonal Precipitation Forecasts. *Journal of Hydrology, 610*, 127896. https://doi.org/10.1016/j.jhydrol.2022.127896

Mao, G., Vogl, S., Laux, P., Wagner, S., & Kunstmann, H. (2015). Stochastic Bias Correction of Dynamically Downscaled Precipitation Fields for Germany Through Copula-Based Integration of Gridded Observation Data. *Hydrology and Earth System Sciences, 19*(4), 1787-1806. https://doi.org/10.5194/hess-19-1787-2015

**Comment:** In Chapter 4.3, "the EC-QM (EC-CNN) forecasts reduce the relative error of the raw forecasts by 12% (20%), 16% (24%) and 9% (21%), respectively, and reduces the relative error of maximum daily flow by 27% (29%), 16% (32%) and 11% (18%)", whether EC-QM or EC-CNN showed the RMSE decreased more or the lead times of 11-20 days than 1-10 days, but we think that the smaller the lead time, the better the results. Can the author give a reasonable explanation?

**Reply:** Thank you for your comment. To clarify, Chapter 4.3 in the original manuscript discusses the relationship between the *improvements* due to statistical downscaling techniques like QM or CNN and the lead time. While it is true that the forecast accuracy generally decreases with longer lead times (as shown in Figures 4 and 7), this relationship may not be the case when examining the *improvements* brought about by QM or CNN for different lead times. For example, many studies have shown that, in specific cases, the improvement from statistical downscaling could be larger for longer lead times and smaller for shorter lead times (Zhang et al., 2023; Lyu et al., 2023; Li et al. 2024).

We will revise the manuscript to clarify this distinction, ensuring that the discussion on the varying impact of QM and CNN with lead time is clear and well-supported by the literature.

**References**

Zhang, T., Liang, Z., Li, W., Wang, J., Hu, Y., & Li, B. (2023). Statistical Post-Processing of Precipitation Forecasts Using Circulation Classifications and Spatiotemporal Deep Neural

Networks. Hydrology and Earth System Sciences, 27(10), 1945-1962. https://doi.org/10.5194/hess-27-1945-2023

Lyu, Y., Zhu, S., Zhi, X., Ji, Y., Fan, Y., & Dong, F. (2023). Improving Subseasonal-To-Seasonal Prediction of Summer Extreme Precipitation Over Southern China Based on a Deep Learning Method. Geophysical Research Letters, 50(24), e2023GL106245. https://doi.org/10.1029/2023GL106245

Li, L., Yun, Z., Liu, Y., Wang, Y., Zhao, W., Kang, Y., & Gao, R. (2024). Improving Categorical and Continuous Accuracy of Precipitation Forecasts by Integrating Empirical Quantile Mapping and Bernoulli-Gamma-Gaussian Distribution. Atmospheric Research, 298, 107133. https://doi.org/10.1016/j.atmosres.2024.107133

**Comment:** Since the article analyses individual cases, in conjunction with Figure 8, please give quantitative indicators to analyse the description.

**Reply:** We agree that quantitative analysis will strengthen the discussion of individual cases. We will include quantitative indicators alongside Figure 8 to provide a more robust analysis of the streamflow forecasts, comparing them across different scenarios.

**Minor Comments:**

**Comment:** L38: "S. Zhu et al., 2020" should be modified.

**Reply:** We will correct the citation formatting in L38 as recommended.

**Comment:** L60: "Wilby, R. L., et al., 2004; Vrac, M., & Friederichs, P., 2015" should be modified.

**Reply:** We will adjust the citation formatting in L60 for consistency and correctness.

**Comment:** L87-88: "For example, Humphrey et al. (2016) achieved improved streamflow forecast skills by combining Bayesian artificial neural networks with traditional models of GR4J." What's the conclusion?

**Reply:** We will clarify the conclusion drawn from Humphrey et al. (2016) in the revised manuscript.

**Comment:** L142-143: "streamflow and flood peak forecasts of XAJ-LSTM and standalone XAJ driven by EC-CNN forecasts are then quantitatively evaluated against those driven by raw and QM-based forecasts using a series of metrics. " Please write the full name for the first occurrence.

**Reply:** We will ensure that the full names of all abbreviations are provided upon their first occurrence in the manuscript.

**Comment:** L262-263: "The results indicate that the daily Nash-Sutcliffe Efficiency (NSE)", abbreviations have already been mentioned.

**Reply:** We will correct this by removing the repeated explanation of the abbreviation.

---

## Author Comment (AC2)

**Response to Reviewer #2**

Thank you for your comments. Below are our replies to each of your comment.

We believe that the planned changes will improve the clarity and significance of our manuscript.

**Comment:** Although QM is a widely applied technique and should be familiar to the general audience of HESS, the reviewer believes that some key information or references may need to be incorporated. Precipitation data is known to be highly skewed, making the selection of the distribution function critically important for the effectiveness of QM. However, such details appear to be missing in the current manuscript.

**Reply:** In our current implementation of QM, a specific distribution function is not required as we use a non-parametric approach. This method directly adjusts the quantiles of the forecasted and observed data without assuming a specific distribution. To be specific, the empirical cumulative distribution functions of observed and forecasted daily precipitation are built respectively, and each percentile of the forecasted data is adjusted to match the corresponding percentile in the observed data. Dry days with a precipitation amount less than 0.1 mm are excluded from the derivation of empirical cumulative distribution functions. Several studies have indicated the effectiveness of this approach in improving the overall precipitation forecasts (Manzanas et al., 2018; Cannon et al., 2015).

We will revise the manuscript to provide details on the implementation of the QM approach and provide the necessary references to support it.

**References**
Manzanas, R., Lucero, A., Weisheimer, A., & Gutiérrez, J. M. (2018). Can bias correction and statistical downscaling methods improve the skill of seasonal precipitation forecasts? Climate dynamics, 50, 1161-1176.
Cannon, A. J., Sobie, S. R., & Murdock, T. Q. (2015). Bias correction of GCM precipitation by quantile mapping: How well do methods preserve changes in quantiles and extremes? Journal of Climate, 28(17), 6938-6959.

**Comment:** Additionally, the reviewer is curious about the specific implementation of QM. Was seasonality or the variation in forecast lead times considered in the QM-based bias removal process? More detailed documentation on this aspect should be included in the manuscript.

**Reply:** In our current implementation, the QM is constructed separately for each lead time to account for forecast bias variations across different lead times. For each lead time, a single model is applied across all months, which is aligned with the structure of the CNN model built in this study. This approach is considered able to more effectively capture biases across different lead times while maintaining a uniform correction across the seasonal cycle.

We will clarify this aspect in the manuscript and provide more details on the implementation of the QM process to ensure a clear understanding of the methodology.

**Comment:** Furthermore, given that QM is primarily designed for bias removal rather than

enhancing the temporal correspondence between forecast time series and observations, the reviewer suggests that the authors also evaluate the resulting forecasts (both precipitation and streamflow) in terms of their bias. Specifically, the overall CDF of precipitation forecasts generated by different statistical downscaling methods should be compared. Given that these downscaled precipitation forecasts eventually run through lumped hydrologic models, CDF of the areal averaged precipitation forecasts is perhaps a good way to demonstrate the bias condition at all percentiles across the study region. While the proposed DL technique improves the predictive skill of S2S precipitation, it would be valuable to see whether it also reduces forecast bias compared to QM.

**Reply:** We agree that while QM primarily addresses bias removal, it would be beneficial to evaluate both the precipitation and streamflow forecasts in terms of bias. In response to your suggestion, we plan to compare the cumulative distribution functions (CDFs) of the areal-averaged precipitation forecasts generated by QM and the proposed deep learning (DL) framework, which could provide a comprehensive view of the bias at all percentiles.

We believe this additional analysis will allow us to assess not only the predictive skill but also the extent to which the DL technique reduces forecast bias compared to QM. We will incorporate this comparison into the revised manuscript to offer a clearer picture of the effectiveness of our approach in reducing forecast bias.

**Comment:** The reviewer feels that the description of the employed statistical downscaling techniques is unclear in general. It appears that the proposed CNN-ResNet generates a single precipitation prediction value while using multiple spatially distributed forecast variables as inputs (Figure 2). If this is indeed the case, the proposed framework seems more like an "upscaling " rather than "downscaling" technique. This also raises questions about how the authors produced the spatially distributed precipitation climatology plot (Figure 6). Additionally, given that CNN-based structures typically produce square-shaped outputs, were any masks applied during the training of the proposed CNN-ResNet?

**Reply:** The CNN incorporating residual blocks, or the CNN-ResNet framework, in our study is indeed a downscaling technique. Specifically, it downscales ECMWF S2S reforecasts from a coarse 1.5-degree resolution to a finer 0.25-degree resolution, which corresponds to the resolution of the CN05.1 observation-based dataset. The CNN uses spatially distributed inputs (e.g., geopotential height, temperature, humidity) from the ECMWF dataset, covering 7×7 1.5-degree coarse grid cells centered on the target 0.25-degree fine grid cell. Due to the square-shaped input structure of the CNN model, some ECMWF data from outside the basin boundary are included in the input. For the outputs, the CNN loops over each fine-resolution grid cell (0.25 degrees) within the basin boundary, generating a high-resolution precipitation forecast for the region of interest. Therefore, no masks are applied during training, as the model is trained to predict each fine-resolution grid cell individually within the basin boundary.

We will clarify this process in the manuscript to avoid any potential confusion about the methodology and ensure that the CNN-ResNet framework is well understood.

**Comment:** Similarly, is QM conducted at each pixel across the study watershed? If so, does the raw spatial resolution of the S2S precipitation forecast match that of the reference precipitation? These questions are particularly relevant considering the employed hydrologic models are lumped. It is

important to clarify for the audience at which specific technical step(s) the spatially distributed forecast variables are converted into area averages.

In general, it is recommended that the entire methodology section be revised to avoid potential confusion and to ensure clarity on the steps involved in the downscaling process.

**Reply:** The raw ECMWF S2S precipitation reforecasts are released at a 1.5-degree resolution, while the reference CN05.1 observation dataset is at a 0.25-degree resolution. To match the forecast resolution with reference dataset resolution, QM is performed by looping over 0.25-degree fine grid cells to establish the empirical cumulative distribution function for each fine grid cell based on its corresponding 1.5-degree coarse grid cell from ECMWF reforecasts. This method is widely used in statistical downscaling to account for the resolution mismatch between coarse forecast models and high-resolution observational data (Gudmundsson et al., 2012).

Given that the hydrologic models employed in this study are lumped, the spatially distributed precipitation forecasts (both from QM and CNN) are converted to area averages after the downscaling step. This averaging occurs within the study area before the precipitation data are input into the hydrologic models.

We agree that these technical steps should be clarified to avoid any confusion. We will revise the methodology section to clearly outline the process, including how the downscaling is conducted at each pixel and how the spatially distributed data are subsequently converted into area averages for use in the lumped hydrologic models.

**References**

Gudmundsson, L., Bremnes, J. B., Haugen, J. E., & Engen-Skaugen, T. (2012). Downscaling RCM precipitation to the station scale using statistical transformations – A comparison of methods. Hydrology and Earth System Sciences, 16(9), 3383-3390. doi:10.5194/hess-16-3383-2012

**Comment:** The reviewer suggests conducting additional seasonal and spatial analysis to better highlight the strengths and weaknesses of the proposed CNN-ResNet downscaling technique. First, the reviewer notes that the study watershed covers a broad area (around 10 degrees, or approximately 1000 km, in both the north-south and east-west directions). This suggests significant spatial and seasonal variability in terms of precipitation generation mechanisms, magnitude, frequency, etc. However, this variability is not discussed in the manuscript, limiting the audience's understanding of the study watershed.

Building on this, it would be interesting to examine whether the proposed framework is equally effective across different seasons and geospatial locations, or if its performance varies. The reviewer believes such an analysis would be crucial in further enhancing the quality of the manuscript. Consequently, it is recommended that the authors evaluate the post-processed precipitation both spatially and seasonally. Since the proposed method is a statistical downscaling technique, it is important to demonstrate its skill over such a large study region. This additional analysis would provide valuable insights into the effectiveness of the method across different conditions.

**Reply:** Thank you for your suggestion. We fully understand the importance of examining spatial and seasonal variability across such a large and diverse study area.

While we agree that spatial analysis is crucial due to the broad area covered by the basin and

its variability in precipitation generation mechanisms, our downscaling and model evaluations were all conducted during the wet season, specifically from May to August. This period is chosen because it accounts for the majority of annual precipitation, which is key to water resources management and flood prevention in the region. Therefore, an evaluation across different seasons may not be the primary focus of this paper. We will revise the manuscript to highlight the focus is on the wet season.

However, we fully agree that conducting a spatial analysis within the wet season, with the geographic variability (e.g., northern plateau versus southern hill regions) taken into account, will provide important insights into the performance of the CNN-ResNet model. This analysis will allow us to assess if the model performs consistently across the different parts of the watershed and help identify any areas where its performance may vary.

A preliminary spatial analysis on the RMSE of EC and EC-CNN forecast for lead times of 1-10, 11-20 and 21-30 days is presented in the Figure R1 below. It can be seen that the EC-CNN improves the forecast skills of the raw ECMWF forecasts over the majority of the basin for all lead times. For example, the RMSE is reduced from 3.4 mm/day to an average of 2.2 mm/day at the northern headwaters of the basin for the lead time of 21-30 days. Similar improvements can also be seen around the southern part of the basin.

[Figure]

**Figure R1**. RMSE of EC and EC-CNN forecasted precipitation for lead times of 1-10, 11-20 and 21-30 days.

We will perform a comprehensive spatial analysis and revise the manuscript accordingly.

**Comment:** If the potential workload is manageable, the reviewer strongly recommends that the authors utilize the entire ensemble of S2S precipitation forecasts from ECMWF in their experiments, rather than focusing on the ensemble means. The primary reason for this suggestion is that neither precipitation forecasts nor the corresponding streamflow predictions can be applied deterministically at a subseasonal timescale due to limited skills at longer forecast lead times.
At this timescale, probabilistic forecasts are typically constructed using multiple predictions (i.e., ensemble forecasts). While the proposed framework appears effective and interesting, the reviewer

believes its full potential can be better demonstrated with a revised experimental design that aligns more closely with real-world needs (i.e., ensemble predictions).

Following this suggestion, the reviewer suggests the authors to incorporate additional probabilistic evaluation metrics, such as CRPS or CRPSS, for a more comprehensive assessment of the framework's performance for both post-processed precipitation forecasts and the corresponding streamflow predictions.

**Reply:** We agree with your suggestion to incorporate ensemble-based predictions, as probabilistic forecasts are more suitable for sub-seasonal timescales due to the inherent uncertainties at longer lead times (Li et al., 2019; Ferranti et al., 2018). To address this, we will use all ensemble members from the ECMWF S2S precipitation reforecasts and build ensemble CNN models to generate probabilistic forecasts. For the probabilistic evaluation, we will apply Continuous Ranked Probability Skill Score (CRPSS) to the ensemble forecasts. This metric is widely used for evaluating ensemble precipitation forecasts and accounting for uncertainties across multiple ensemble members (Bremnes, 2020).

We will revise the manuscript to present probabilistic forecasting and its benefits for improving the predictability of precipitation and streamflow on sub-seasonal timescales. Thank you again for this suggestion.

**References**

Bremnes, J. B. (2020). Ensemble postprocessing using quantile function regression based on neural networks and Bernstein polynomials. Monthly Weather Review, 148(1), 403-414.

Li, W., Pan, B., Xia, J., and Duan, Q. (2021). Convolutional neural network-based statistical post-processing of ensemble precipitation forecasts. Journal of hydrology, 605, 127301.

Ferranti, L., Corti, S., & Janousek, M. (2018). Flow-dependent verification of the ECMWF ensemble over the Euro-Atlantic sector. Quarterly Journal of the Royal Meteorological Society, 144(712), 317-326.

**Comment:** Lien 126: What is the naive spatial resolution of the collected S2S precipitation forecasts from ECMWF?

**Reply:** The S2S precipitation reforecasts from ECMWF collected in this study are with a spatial resolution of 1.5 degrees. We will clarify this point in the revised manuscript to ensure the spatial resolution of the input data is clearly understood.

**Comment:** Line 142: EC-CNN is referenced here for the first time in the manuscript, but without a clear explanation.

**Reply:** We will provide a more detailed description when first introducing EC-CNN, which refers to the statistically downscaled ECMWF S2S reforecasts using the proposed CNN framework.

**Comment:** Line 250: It seems a standardized metric is employed here (i.e., NSE) to evaluate the hydrologic model calibration. The reviewer wonders why switch to RMSE and other metrics for later streamflow predictive skill evaluation? While RMSE is a widely applied metric in many fields, standardized metrics such as NSE and KGE might be more familiar to researchers in the hydrology community.

**Reply:** It is true that the hydrologic model calibration is evaluated using the Nash-Sutcliffe Efficiency (NSE). For consistency and to align with hydrologic model evaluation metrics, we agree that it would be appropriate to add NSE for the later streamflow predictive skill evaluation as well, and we will revise the manuscript accordingly.

**Comment:** Line 354: Perhaps "forecast issue date" is more appropriate for the titles of different panels in Figure 8. Also, it would be interesting to see these examples where the proposed framework delivers more accurate streamflow predictions. Overall skill evaluation would still be more informative in general. Perhaps these figures could be included in the supplementary material so that previous suggested additional evaluation and analysis could be included in the main manuscript.

**Reply:** We agree that using 'forecast issue date' would be more appropriate for the titles of different panels in Figure 8. We will update the figure accordingly in the revised manuscript.

Additionally, we will follow your recommendation to prioritize more comprehensive evaluation and analysis in the main manuscript, while moving very detailed figures and additional examples to the supplementary material. We believe this will allow for a clearer focus on the overall skill evaluation in the main text, while still providing valuable examples for interested readers.

---

## Author Response (AR1)

**Response to the Reviewers of "Deep learning based sub-seasonal precipitation and streamflow forecasting over the source region of the Yangtze River" [Paper #HESS-2024-212]**

In this response letter, comments by the Reviewers are in black, and our replies are in blue. The line references in our replies refer to the PDF clean version of the revised manuscript.

**We highlight the major changes as follows:**

1) In response to Reviewer #1, we have revised the introduction to include recent studies on the progress made in statistical downscaling methods, particularly when combined with deep learning techniques.

2) In response to Reviewer #1 & #2, we have included spatial analyses and relevant plots of precipitation forecast errors in the revised manuscript to exhibit the effectiveness of the proposed DL-based framework in improving sub-seasonal precipitation forecasts in different parts of the study region.

3) In response to Reviewer #1 & #2, we have included the detailed implementation procedure of QM for better clarity and a short discussion on the limitations of QM in the revised manuscript.

4) In response to Reviewer #2, we have thoroughly revised the methodology section and provided more details on the implementation of the CNN model, including the resolution, downscaling procedure, and input and output structure, for better clarity.

5) In response to Reviewer #2, we have added analysis on cumulative distribution function of different precipitation forecasts.

6) In response to the suggestion of Reviewer #2 to include ensemble forecasts, we have employed 10 ensemble members of ECMWF raw forecasts to build ensemble downscaling models in the revised manuscript. CRPS are introduced to evaluate the skills of sub-seasonal ensemble precipitation and streamflow forecasts.

We hope our revision well address the comments from the editor and reviewers.

**Response to Reviewer #1**

Thank you for your comments. Below are our replies to each of your comment.

**Major Comments:**

**Comment:** In introduction, L45-70 introduce the advantages and disadvantages of dynamic downscaling and statistical downscaling, which are well known to us, and just to illustrate a few deep learning methods. Failure to highlight the research focus of this paper, i.e., the progress of research on traditional statistical downscaling and statistical downscaling combined with deep learning.

**Reply:** We have revised the introduction to include more recent studies on the progress made in statistical downscaling methods, particularly when combined with deep learning techniques.

The following text in the Introduction has been modified:

Line 59-78: On the other hand, forecasting weather and predicting climate using machine learning, especially deep learning (DL), has recently become a hot topic. A common approach for this purpose is to use preceding predictors from observational or reanalysis data to forecast subsequent predictands (Weyn et al., 2021; Xie et al., 2023; Ham et al., 2019). An alternative method involves postprocessing dynamical forecasts. For instance, Cho et al. (2020) applied machine learning techniques, including random forests and support vector machines, to develop statistical relationships for temperature adjustments. Similarly, Kim et al. (2021) utilized Long Short-Term Memory (LSTM) networks to correct bias in the amplitude and phase of the Madden–Julian Oscillation. More recently, deep learning models such as convolutional neural networks (CNN) have been reported able to more effectively reduce the total bias of meteorological forecasts due to their ability to learn multi-dimensional representations of data features (Vandal et al., 2019; Sachindra et al., 2018; Ning et al., 2024). For example, Lagerquist et al. (2019) used a CNN to identify fronts in gridded data for spatially explicit prediction of synoptic-scale fronts.

Despite general improvements of forecasts, these DL-based models tend to smooth the extreme precipitation at sub-seasonal scales (Baño-Medina et al., 2021; Kim et al., 2022), likely due to insufficient heavy precipitation samples (Chen et al., 2022). Many studies have since introduced more recent variants of CNNs including the U-shaped U-

Net (Han et al, 2021; Horat and Lerch, 2024; Ni et al., 2023) and SmaAt-UNet (Li et al., 2024), or coupled standard CNNs with different structures, such as Auto-Encoder (Ling et al., 2022) and Transformer (Ling et al., 2024). In particular, the residual network, ResNet, has been introduced in sub-seasonal forecast correction, which shows the potential of mitigating the vanishing gradient issue by introducing the residual paths (Jin et al 2022; Nie et al., 2024). Others have attempted to introduce specialized loss functions to balance heavy and light rains, such as the exponentially weighted mean squared error (Ebert-Uphoff et al., 2020) and Dice loss (You et al., 2022). However, these new developments have not been sufficiently examined for sub-seasonal forecasts.

**Comment:** I do not understand the role of L70-75, it seems to describe weather prediction models like Pangu and GraphCast which can achieve entirely through deep learning have demonstrated the potential to achieve forecast skills comparable to state-of-the-art numerical weather prediction systems. Is this relevant to the research in this paper?

**Reply:** The mention of Pangu and GraphCast was intended to introduce the state-of-the-art *fully DL-based models* for weather prediction from a broader context. To bridge the gap between *fully DL-based models* and *statistically post-processing DL models* (which is the focus of our paper), we have revised the following text to make this paragraph directly relevant to our study.

Line 79-85: Other state-of-the-art forms of deep learning for weather forecasts include fully DL-based models, such as Pangu (Bi et al., 2023) and GraphCast (Lam et al., 2023), which are reported able to achieve forecast skills comparable to numerical weather prediction systems. While these models may appear quite different from statistical post-processing deep learning models, some argue that these models act more as post-processing tools rather than realistic simulators of the atmosphere due to the lack of physical fidelity and consistency (Bonavita et al., 2024). Although not the primary focus of this paper, this calls attention to the scientific community to critically evaluate and differentiate between the capabilities and applications of fully DL-based models and DL models designed for post-processing.

**Comment:** About result, some of the conclusions seem too brief. L255-260 have no results about the relative error stands, the mean absolute error and relative error of simulated maximum daily flow.

**Reply:** We have included more metrics in the paragraph. The following text has been added to the revised manuscript:

Line 295-297: The relative error of streamflow is 1.0% and 2.6% during the calibration and validation period. The mean absolute error and relative error of simulated maximum daily flow are 844 $m^3$/s and 17.0% during the calibration period, and are 379 $m^3$/s and 7.9% during the validation period, respectively.

**Comment:** The author writes "while the RMSE of EC-QM forecasts sees a relatively steady reduction over all lead times"(L280-285), but from the Fig.4, we can clearly find that EC-QM showed more RMSE compared to EC for the lead times of 23-24 days, 27 days. The author can explain the reason?

**Reply:** We agree with the Reviewer that the QM method does not reduce the RMSE in an exactly steady manner with lead time. In particular, for several lead time ranges, QM-based forecasts show almost equal or even more RMSEs to those of raw EC forecasts. Such finding is also observed by some of the recent studies using QM for statistical downscaling precipitation. For example, Li et al. (2023), Huang et al. (2022), and Mao et al. (2015) show that, while QM is generally effective in adjusting model bias towards observations, it does not always lead to improvements of the forecast.

One plausible reason could be the limited applicability of the relatively simple QM method. Specifically, QM primarily adjusts the distribution of forecasted values towards the distribution of historical observations, with no account of the atmospheric conditions associated with those forecasts. However, physics-based numerical weather predictions involve complex and nonlinear errors that QM may not be able to fully correct. Another reason could be that QM does not explicitly account for errors in the spatial dependencies of precipitation field, which can also contribute to the increased RMSE in specific lead times.

The following text and figure have been added to the revised manuscript:

Line 452-460: It is noted that, for several lead time ranges, EC-QM forecasts show no improvements of RMSEs compared to those of raw EC forecasts. This was also

observed by some of the recent studies using QM for statistical downscaling precipitation. For example, Li et al. (2023), Huang et al. (2022), and Mao et al. (2015) show that, while QM is generally effective in adjusting model bias towards observations, it does not always lead to improvements of the forecast accuracy. One plausible reason could be the limited applicability of the relatively simple QM method. Specifically, QM primarily adjusts the distribution of forecasted values towards the distribution of historical observations, with no account of the atmospheric conditions associated with those forecasts. However, physics-based numerical weather predictions involve complex and nonlinear errors that QM may not be able to fully correct.

**References**

Li, X., Wu, H., Nanding, N., Chen, S., Hu, Y., & Li, L. (2023). Statistical Bias Correction of Precipitation Forecasts Based on Quantile Mapping on the Sub-Seasonal to Seasonal Scale. *Remote Sensing, 15*(7), 1743. https://doi.org/10.3390/rs15071743

Huang, Z., Zhao, T., Xu, W., Cai, H., Wang, J., Zhang, Y., Liu, Z., Tian, Y., Yan, D., & Chen, X. (2022). A Seven-Parameter Bernoulli-Gamma-Gaussian Model to Calibrate Subseasonal to Seasonal Precipitation Forecasts. *Journal of Hydrology, 610*, 127896. https://doi.org/10.1016/j.jhydrol.2022.127896

Mao, G., Vogl, S., Laux, P., Wagner, S., & Kunstmann, H. (2015). Stochastic Bias Correction of Dynamically Downscaled Precipitation Fields for Germany Through Copula-Based Integration of Gridded Observation Data. *Hydrology and Earth System Sciences, 19*(4), 1787-1806. https://doi.org/10.5194/hess-19-1787-2015

**Comment:** And please draw the spatial distribution of the bias of EC-CNN, EC-QM and EC because it mentioned in Line 310.

**Reply:** Done. The following text and figure have been added to the revised manuscript:

Line 343-352: To investigate the spatial characteristics of precipitation forecasts, Figure 5 presents the spatial distribution on the RMSE of EC and EC-CNN forecast for lead times of 1-10, 11-20 and 21-30 days. It is clear that the EC-CNN improves the forecast skills of the raw ECMWF forecasts over the majority of the SR for all lead times. For example, the RMSE is reduced from 3-5 mm/day for EC forecasts to 1-2 mm/day for EC-CNN forecasts at the northern sub-basin for all lead times. Similar improvements can also be seen around the southern part of the basin, for example the southmost part

of SR sees RMSE over 10 mm/day for EC forecasts at lead times of 11-30 days but reduces to 6-7 mm/days for EC-CNN forecasts. In addition, by comparing Fig. R1b and Fig. R1c it can be seen that the EC-CNN shows larger improvements than EC-QM across the SR for all lead times. The above results indicate that EC-CNN not only improves the raw forecasts temporally, but also enhances their spatial accuracy across various regions of the SR. This basin-wise improvement allows for more reliable predictions across diverse hydrological zones within the SR, which could further benefit the hydrologic modelling.

[Figure]

Figure 5. The spatial distribution of RMSE for the ensemble means of (a) EC forecasts, (b) EC-QM, and (c) EC-CNN forecasts averaged over lead times of 1-10 days, 11-20 days, and 21-30 days during the test period.

**Comment:** In Chapter 4.3, "the EC-QM (EC-CNN) forecasts reduce the relative error of the raw forecasts by 12% (20%), 16% (24%) and 9% (21%), respectively, and reduces the relative error of maximum daily flow by 27% (29%), 16% (32%) and 11% (18%)", whether EC-QM or EC-CNN showed the RMSE decreased more or the lead times of 11-20 days than 1-10 days, but we think that the smaller the lead time, the better the results. Can the author give a reasonable explanation?

**Reply:** Thank you for your comment. To clarify, Section 4.3 discusses the relationship between the *improvements* due to statistical downscaling techniques like QM or CNN and the lead time. While it is true that the forecast accuracy generally decreases with longer lead times (as shown in Figures 4 and 8 in the revised manuscript), this relationship may not be the case when examining the *improvements* brought about by QM or CNN for different lead times. For example, many studies have shown that, in specific cases, the improvement from statistical downscaling could be larger for longer lead times and smaller for shorter lead times (Zhang et al., 2023; Lyu et al., 2023; Li et al. 2024).

The following text has been added in the revised manuscript:

Line 391-392: It is noted that the improvements brought by EC-QM and EC-CNN are larger in lead times of 11-20 days than in 1-10 days, which was also observed by Zhang et al. (2023), Lyu et al. (2023), and Li et al. (2024) in specific cases.

**References**

Zhang, T., Liang, Z., Li, W., Wang, J., Hu, Y., & Li, B. (2023). Statistical Post-Processing of Precipitation Forecasts Using Circulation Classifications and Spatiotemporal Deep Neural Networks. Hydrology and Earth System Sciences, 27(10), 1945-1962. https://doi.org/10.5194/hess-27-1945-2023

Lyu, Y., Zhu, S., Zhi, X., Ji, Y., Fan, Y., & Dong, F. (2023). Improving Subseasonal-To-Seasonal Prediction of Summer Extreme Precipitation Over Southern China Based on a Deep Learning Method. Geophysical Research Letters, 50(24), e2023GL106245. https://doi.org/10.1029/2023GL106245

Li, L., Yun, Z., Liu, Y., Wang, Y., Zhao, W., Kang, Y., & Gao, R. (2024). Improving Categorical and Continuous Accuracy of Precipitation Forecasts by Integrating Empirical Quantile Mapping and Bernoulli-Gamma-Gaussian Distribution. Atmospheric Research, 298, 107133. https://doi.org/10.1016/j.atmosres.2024.107133

**Comment:** Since the article analyses individual cases, in conjunction with Figure 8, please give quantitative indicators to analyse the description.

**Reply:** Done. We have added the relative error metric in each subplot of streamflow forecasts for straightforward evaluation. Relative error is selected because it is a

straightforward metric to reflect the amount of water resources in the future, which plays an important role in water management downstream. The relevant description has also been modified to include quantitative metrics of different streamflow forecasts.

The following text has been modified in the revised manuscript:

Line 416-428: Figure 9 presents examples of XAJ-LSTM streamflow ensemble forecasts issued on different dates, with the RE of total forecast flow presented in each subplot. In most cases, the EC-CNN forecasts can reduce the streamflow bias in a more flexible manner than the EC-QM forecasts, resulting in more accurate overall streamflow predictions across different dates. For example, the CNN model decreases (increases) the EC precipitation and hence the forecast streamflow issued on June 06, 2017 (August 08, 2019), which improves the forecast skills in both cases. The EC-CNN reduces the relative error from 22.1% of raw EC forecasts to -2.5% for the 30-day streamflow forecast issued on June 06, 2017, and reduces the relative error from -21.5% of raw EC forecasts to -9.2% for the 30-day streamflow forecast issued on August 08, 2019. On the other hand, the QM reduces the precipitation and hence the streamflow forecasts for both dates, which reduces the relative error to 11.0% on June 2017 but increases the relative error to -27.3% on August 2019. Similarly, for the forecast issued on Aug 11, 2017, EC-CNN decreases the EC precipitation for lead times of 1-20 days and increases it for lead times of 21-30 days, which alleviates the streamflow overestimation on late August and underestimation on early September. However, EC-QM consistently predicts lower precipitation and hence streamflow compared to EC forecasts at all lead times, worsening the underestimation in early September.

[Figure]

Figure 9. Examples of sub-seasonal XAJ-LSTM streamflow forecasts for a lead time of 30 days driven by EC, EC-QM, and ensemble EC-CNN precipitation forecasts. The relative error of total forecast flow is shown in each subplot.

**Minor Comments:**

**Comment:** L38: "S. Zhu et al., 2020" should be modified.

**Reply:** Done.

**Comment:** L60: "Wilby, R. L., et al., 2004; Vrac, M., & Friederichs, P., 2015" should be modified.

**Reply:** Done.

**Comment:** L87-88: "For example, Humphrey et al. (2016) achieved improved streamflow forecast skills by combining Bayesian artificial neural networks with traditional models of GR4J." What's the conclusion?

**Reply:** We have added the conclusion as follows.

Line 97-98: Humphrey et al. (2016) combined a Bayesian neural network (BNN) with the traditional GR4J model and achieved improved forecast accuracy compared to using either the BNN or GR4J alone.

**Comment:** L142-143: "streamflow and flood peak forecasts of XAJ-LSTM and standalone XAJ driven by EC-CNN forecasts are then quantitatively evaluated against those driven by raw and QM-based forecasts using a series of metrics. " Please write the full name for the first occurrence.

**Reply:** Done.

Line 151-155: We first employ all 10 ensemble members from the ECMWF S2S gridded sub-seasonal precipitation reforecast dataset for the next 30 days as raw forecasts, denoted as EC. An ensemble of enhanced CNN models with ResNet blocks and a specialized loss function is established to statistically downscale and bias correct each ensemble member of the 1.5° EC raw precipitation forecasts to 0.25° grid resolution, with its post-processed forecast denoted EC-CNN (Section 3.2.1). The quantile mapping (QM) serves as a benchmark for comparison, with its post-processed forecast denoted EC-QM (Section 3.2.2).

Line 159-163: The first hydrologic model is a standalone XAJ model (Section 3.4.1), and the second model is a hybrid model that integrates the conceptual XAJ model and the LSTM (hereinafter XAJ-LSTM) (Sections 3.4.2 and 3.4.3). The streamflow and flood forecasts of XAJ-LSTM and standalone XAJ driven by EC-CNN forecasts are then quantitatively evaluated against those driven by EC and EC-QM forecasts.

**Comment:** L262-263: "The results indicate that the daily Nash-Sutcliffe Efficiency (NSE)", abbreviations have already been mentioned.

**Reply:** This has been corrected.

**Response to Reviewer #2**

Thank you for your comments. Below are our replies to each of your comment.

**Comment:** Although QM is a widely applied technique and should be familiar to the general audience of HESS, the reviewer believes that some key information or references may need to be incorporated. Precipitation data is known to be highly skewed, making the selection of the distribution function critically important for the effectiveness of QM. However, such details appear to be missing in the current manuscript.

**Reply:** In the current implementation of QM, a specific distribution function is not required as we use a non-parametric approach. This method directly adjusts the quantiles of the forecasted and observed data without assuming a specific distribution. We have added detailed information on the non-parametric implementation of QM in the revised manuscript, and the following text has been added in the revised manuscript:

Line 227-234: We implement QM using a non-parametric approach that adjusts the quantiles of the forecasted and observed data without assuming a specific distribution. Specifically, the empirical cumulative distribution functions (CDFs) of observed and forecasted daily precipitation are built respectively, and each percentile of the forecasted data is adjusted to match the corresponding percentile in the observed data. Dry days with a precipitation amount less than 0.1 mm are excluded from the derivation of CDFs (Gudmundsson et al., 2012). To match the 1.5° forecast resolution with the 0.25° reference dataset resolution, the empirical CDFs are established separately for each 0.25° grid from the corresponding 1.5° forecast grid cell. Manzanas et al (2018), Cannon et al. (2015) and other studies have indicated the effectiveness of this implementation in improving the overall precipitation forecasts.

**References**

Manzanas, R., Lucero, A., Weisheimer, A., & Gutiérrez, J. M. (2018). Can bias correction and statistical downscaling methods improve the skill of seasonal precipitation forecasts? Climate dynamics, 50, 1161-1176.

Cannon, A. J., Sobie, S. R., & Murdock, T. Q. (2015). Bias correction of GCM precipitation by quantile mapping: How well do methods preserve changes in quantiles and extremes? Journal of Climate, 28(17), 6938-6959.

Gudmundsson, L., Bremnes, J. B., Haugen, J. E., & Engen-Skaugen, T. (2012). Downscaling RCM precipitation to the station scale using statistical transformations – A comparison of methods. Hydrology and Earth System Sciences, 16(9), 3383-3390. doi:10.5194/hess-16-3383-2012

**Comment:** Additionally, the reviewer is curious about the specific implementation of QM. Was seasonality or the variation in forecast lead times considered in the QM-based bias removal process? More detailed documentation on this aspect should be included in the manuscript.

**Reply:** The QM is constructed separately for each lead time to account for forecast bias variations across different lead times. For each lead time, a single model is applied across all months, which is aligned with the structure of the CNN model built in this study. This approach is considered able to more effectively capture biases across different lead times while maintaining a uniform correction across the seasonal cycle.

The following text has been added in the revised manuscript:

Line 238-240: The QM is constructed separately for each lead time to account for forecast bias variations across different lead times. For each lead time, a single model is applied across all months, which is aligned with the structure of the CNN model built in this study.

**Comment:** Furthermore, given that QM is primarily designed for bias removal rather than enhancing the temporal correspondence between forecast time series and observations, the reviewer suggests that the authors also evaluate the resulting forecasts (both precipitation and streamflow) in terms of their bias. Specifically, the overall CDF of precipitation forecasts generated by different statistical downscaling methods should be compared. Given that these downscaled precipitation forecasts eventually run through lumped hydrologic models, CDF of the areal averaged precipitation forecasts is perhaps a good way to demonstrate the bias condition at all percentiles across the study region. While the proposed DL technique improves the predictive skill of S2S precipitation, it would be valuable to see whether it also reduces forecast bias compared to QM.

**Reply:** In response to your suggestion, we calculate and compare the cumulative distribution function of the areal-averaged precipitation forecasts of the study region

generated by EC-QM and EC-CNN in the test period. The following analysis text has been added in the revised manuscript:

Line 356-367: Figure 6 presents cumulative distribution function (CDFs) of EC, EC-QM and EC-CNN forecasted precipitation averaged over the SR across the lead times. Notably, the EC-QM forecast consistently aligns well with the observed CDF across all lead times, which reflects the intended purpose of QM to match the empirical distribution through quantile mapping. This result demonstrates EC-QM is good at correcting the raw EC forecast to follow the observed distribution closely, despite an overall large RMSE and RE compared to EC-CNN.

The EC-CNN forecast shows improvement over the EC forecasts and EC-QM forecasts by better approximating the observed CDF for the first 15 days. However, as compared to EC and EC-QM, the EC-CNN begins to deviate more significantly from the observed CDF as lead times increase. Specifically, the EC-CNN forecast appears to concentrate around medium precipitation values for 16-30 days, which underestimates the frequency of both lighter and heavier precipitation events. This pattern suggests that, while EC-CNN improves the overall accuracy of light and heavy rains of the raw EC forecasts at all lead times (Figure 4), the distributional accuracy, particularly for extreme precipitation events, may be compromised over extended lead times. A discussion on the possible cause of biases in CDF for different forecasts is provided in Section 5.4.

[Figure]

**Figure 6.** The cumulative distribution function (CDF) of areal-averaged precipitation for observed precipitation, the ensemble means of EC, EC-QM, and EC-CNN forecasts at different lead times during the test period.

Line 519-531 of Section 5.4: One limitation of the proposed EC-CNN model is its reduced accuracy in capturing the cumulative distribution function (CDF) over lead times extending beyond 15 days. While EC-CNN outperforms in terms of RMSE and RE for both light and heavy rainfall events (Figure 4), its CDF deviates from observed patterns for these longer lead times. In contrast, EC and EC-QM tend to align more closely with observed CDFs at these lead times. We attribute this to the following possible reason.

Over extended lead times (>15 days), extreme precipitation values of EC and EC-QM forecasts tend to be more incorrectly assigned to specific time, leading to, for instance, storm-level precipitation predictions on dry days or nearly zero precipitation predictions on storm days. While these misplaced extremes inflate daily errors, they still allow the overall CDF of the forecast to retain both high and low ends of the precipitation distribution. The EC-CNN model, however, is designed to minimize daily errors by adjusting outliers and bringing exaggerated values closer to moderate levels, thereby reducing large forecast errors. For example, when EC forecasts a high

precipitation event on a dry day, EC-CNN mitigates this by lowering the extreme to a more typical value. While this adjustment helps decrease RMSE and RE, it inadvertently compresses the distribution toward the center by reducing the frequency of both extreme high and low precipitation values. This approach limits the model ability to capture the full distribution, resulting in a CDF that is overly concentrated around moderate values.

**Comment:** The reviewer feels that the description of the employed statistical downscaling techniques is unclear in general. It appears that the proposed CNN-ResNet generates a single precipitation prediction value while using multiple spatially distributed forecast variables as inputs (Figure 2). If this is indeed the case, the proposed framework seems more like an "upscaling " rather than "downscaling" technique. This also raises questions about how the authors produced the spatially distributed precipitation climatology plot (Figure 6). Additionally, given that CNN-based structures typically produce square-shaped outputs, were any masks applied during the training of the proposed CNN-ResNet?

**Reply:** We agree with you that additional details are needed in the description of methods.

The CNN incorporating ResNet blocks is indeed a downscaling technique in our study. Specifically, it downscales ECMWF S2S ensemble reforecasts from a coarse 1.5° resolution to a finer 0.25° resolution, which corresponds to the resolution of the CN05.1 observation-based dataset. The CNN model takes spatially distributed inputs of 19 predictors, such as geopotential height, temperature, and humidity, from the ECMWF dataset. These inputs cover a 3×3 area of coarse grid cells at 1.5° resolution, centered around the target fine grid cell at 0.25° resolution. The predicted precipitation has a spatial resolution of 0.25 degrees and is areal-averaged to the two sub-basins of the SR before being input into the hydrologic model. Due to the square-shaped input structure of the CNN model, some ECMWF data from outside the basin boundary are included in the input. For the outputs, the CNN loops over each fine-resolution grid cell (0.25 degrees) within the basin boundary, generating a high-resolution precipitation forecast for the study basin. Therefore, no masks are applied during training, as the model is trained to predict each fine-resolution grid cell individually within the basin boundary.

These details are added to Section 3.2.1:

Line 173-181: Specifically, it downscales a total of 10 ensemble members of the ECMWF S2S reforecasts from a 1.5° resolution to a 0.25° resolution, using the CN05.1 reference precipitation dataset. The model takes spatially distributed inputs of 19 predictors, including the surface elevation, convective precipitation, total precipitation at the surface level, and U/V wind components, specific humidity, temperature, and geopotential height at 200/500/850 hPa pressure levels, from ECMWF forecasts. These inputs cover a 3×3 area of coarse grid cells at 1.5° resolution, centered around the target fine grid cell at 0.25° resolution. Due to the square-shaped input structure of the CNN model, some ECMWF data from outside the basin boundary are included in the input. For the outputs, the predictand is the daily precipitation at a spatial resolution of 0.25°, and the CNN loops over each fine-resolution grid cell (0.25°) within the basin boundary, thereby generating a high-resolution precipitation forecast for the entire SR.

**Comment:** Similarly, is QM conducted at each pixel across the study watershed? If so, does the raw spatial resolution of the S2S precipitation forecast match that of the reference precipitation? These questions are particularly relevant considering the employed hydrologic models are lumped. It is important to clarify for the audience at which specific technical step(s) the spatially distributed forecast variables are converted into area averages.

**Reply:** The raw ECMWF S2S precipitation reforecasts are released at a 1.5° resolution, while the reference CN05.1 observation dataset is at a 0.25° resolution. To match the forecast resolution with reference dataset resolution, QM is performed by looping over 0.25° fine grid cells to establish the empirical cumulative distribution function for each fine grid cell based on its corresponding 1.5° coarse grid cell from ECMWF reforecasts. Given that the hydrologic models employed in this study are lumped, the spatially distributed precipitation forecasts (both from QM and CNN) are converted to area averages of two sub-basins of the SR after the downscaling step.

The following text has been added or modified in the revised manuscript:

Line 231-232: To match the 1.5° forecast resolution with the 0.25° reference dataset resolution, the empirical CDFs are established separately for each 0.25° grid from the corresponding 1.5° forecast grid cell.

Line 158-159: All these gridded forecasts are areal-averaged over the two sub-basins of

the SR (Fig. 1) before being input to the lumped hydrologic models.

**Comment:** In general, it is recommended that the entire methodology section be revised to avoid potential confusion and to ensure clarity on the steps involved in the downscaling process.

**Reply:** We have revised the methodology section to clearly outline this process. Please see above replies.

**Comment:** The reviewer suggests conducting additional seasonal and spatial analysis to better highlight the strengths and weaknesses of the proposed CNN-ResNet downscaling technique. First, the reviewer notes that the study watershed covers a broad area (around 10 degrees, or approximately 1000 km, in both the north-south and east-west directions). This suggests significant spatial and seasonal variability in terms of precipitation generation mechanisms, magnitude, frequency, etc. However, this variability is not discussed in the manuscript, limiting the audience's understanding of the study watershed.

Building on this, it would be interesting to examine whether the proposed framework is equally effective across different seasons and geospatial locations, or if its performance varies. The reviewer believes such an analysis would be crucial in further enhancing the quality of the manuscript. Consequently, it is recommended that the authors evaluate the post-processed precipitation both spatially and seasonally. Since the proposed method is a statistical downscaling technique, it is important to demonstrate its skill over such a large study region. This additional analysis would provide valuable insights into the effectiveness of the method across different conditions.

**Reply:** Thank you for your suggestion. We fully understand the importance of examining spatial variability across such a large and diverse study area.

In terms of forecasts in different seasons, our model implementation and evaluations were all conducted during the wet season, specifically from May to August. This period is chosen because it accounts for the majority of annual precipitation, which is key to water resources management and flood prevention in the region. Therefore, an evaluation across different seasons may not be the primary focus of this paper. We have

revised the manuscript to highlight the focus is on the wet season, and the following text has been modified in the revised manuscript:

Line 104-105: Aiming at enhancing sub-seasonal hydrometeorological forecasts for the wet season in this area, we start by addressing the following questions.

Line 149-151: The presented sub-seasonal hydrometeorological forecasting framework aims to improve the daily precipitation forecasts and the corresponding streamflow forecasts at the Shigu hydrologic station during the wet season (May to August) for a lead time up to 30 days.

On the other hand, we fully agree that conducting a spatial analysis within the wet season can provide important insights into the performance of the deep learning model. Therefore, we carried out a spatial analysis on the RMSE of EC, EC-QM and EC-CNN forecasts, and the following text and figure have been added to the revised manuscript:

Line 343-352: To investigate the spatial characteristics of precipitation forecasts, Figure 5 presents the spatial distribution on the RMSE of EC and EC-CNN forecast for lead times of 1-10, 11-20 and 21-30 days. It is clear that the EC-CNN improves the forecast skill of the raw ECMWF forecasts over the majority of the SR for all lead times. For example, the RMSE is reduced from 3-5 mm/day for EC forecasts to 1-2 mm/day for EC-CNN forecasts at the northern sub-basin for all lead times. Similar improvements can also be seen around the southern part of the basin, for example the southernmost part of SR sees RMSE over 10 mm/day for EC forecasts at lead times of 11-30 days but reduces to 6-7 mm/days for EC-CNN forecasts. In addition, by comparing Fig. R1b and Fig. R1c it can be seen that the EC-CNN shows larger improvements than EC-QM across the SR for all lead times. The above results indicate that EC-CNN not only improves the raw forecasts temporally, but also enhances their spatial accuracy across various regions of the SR. This basin-wise improvement allows for more reliable predictions across diverse hydrological zones within the SR, which could further benefit the hydrologic modelling.

[Figure]

Figure 5. The spatial distribution of RMSE for the ensemble means of (a) EC forecasts, (b) EC-QM, and (c) EC-CNN forecasts averaged over lead times of 1-10 days, 11-20 days, and 21-30 days during the test period.

**Comment:** If the potential workload is manageable, the reviewer strongly recommends that the authors utilize the entire ensemble of S2S precipitation forecasts from ECMWF in their experiments, rather than focusing on the ensemble means. The primary reason for this suggestion is that neither precipitation forecasts nor the corresponding streamflow predictions can be applied deterministically at a sub-seasonal timescale due to limited skills at longer forecast lead times.

At this timescale, probabilistic forecasts are typically constructed using multiple predictions (i.e., ensemble forecasts). While the proposed framework appears effective and interesting, the reviewer believes its full potential can be better demonstrated with a revised experimental design that aligns more closely with real-world needs (i.e., ensemble predictions).

Following this suggestion, the reviewer suggests the authors to incorporate additional

probabilistic evaluation metrics, such as CRPS or CRPSS, for a more comprehensive assessment of the framework's performance for both post-processed precipitation forecasts and the corresponding streamflow predictions.

**Reply:** We agree with your suggestion to incorporate ensemble-based predictions, as they are more suitable for sub-seasonal timescales due to the inherent uncertainties at longer lead times (Ferranti et al., 2018). To this end, we employ all 10 ensemble members from the ECMWF S2S precipitation reforecasts and build ensemble QM and CNN models to generate ensemble forecasts. These ensemble precipitation forecasts are then input to hydrologic models for ensemble streamflow forecasting. For the evaluation of ensemble precipitation and streamflow forecasting, we applied Continuous Ranked Probability Skill (CRPS) to the ensemble forecasts. This metric is widely used for evaluating ensemble precipitation forecasts and accounting for uncertainties across multiple ensemble members (Bremnes, 2020).

The following text of evaluation of ensemble precipitation forecasts has been added in the revised manuscript:

Line 371-380: Figure 7 presents the CRPS of EC, EC-QM and EC-CNN forecasted precipitation averaged over the SR across the lead times. CRPS evaluates how close the ensemble forecast distribution is to the observed value, and a value close to zero means a better ensemble forecast. As can be seen in the figure, the CRPS for EC is around 1.1 mm/day at lead times of 1-10 days and increases to around 1.4 mm/day at lead times of 11-30 days. The EC-QM reduces the CRPS by an average of around 0.1 mm/day at all lead times, indicating an improvement in the probabilistic calibration and sharpness of the ensemble forecasts. The EC-CNN further reduces the CRPS for most of the lead times as compared to the EC-QM, especially for the first 5 days where the CRPS is 0.4 mm/day lower than EC forecasts and 0.2 mm/day lower than EC-QM forecasts. This shows that the EC-CNN has an enhanced capability of representing the range of possible outcomes and improving the overall reliability in probabilistic forecasting. Such an advantage also offers better decision-making insights under uncertainty, which is favorable for risk management and planning across various time horizons.

[Figure]

Figure 7. The CRPS of areal-averaged precipitation for the EC, EC-QM, and EC-CNN ensemble forecasts at different lead times during the test period. Error bar represents the 25th-75th percentile interval.

The following text of evaluation of ensemble streamflow forecasts has been added in the revised manuscript:

Line 433-441: Figure 10 presents the CRPS of EC, EC-QM and EC-CNN driven XAJ-LSTM streamflow forecasts at Shigu across the lead times. The CRPS for EC is around 120 m3/s at lead times of 1-5 days and increases rapidly to around 300 $m^3$/s at lead times of 6-10 days, and further to 500 $m^3$/s at lead times of 21-30 days. The EC-QM reduces the CRPS by an average of around 65 $m^3$/s at all lead times, and the EC-CNN further reduces the CRPS for most of the lead times as compared to the EC-QM, especially for the lead times of 6-20 days where the CRPS is about 60 $m^3$/s lower than EC forecasts and 25 $m^3$/s lower than EC-QM forecasts. However, for lead times of 26-30 days, the CRPS of EC-CNN is slightly larger than that of EC-QM, indicating the advantage of EC-CNN in ensemble forecasting is not evident for extended forecast lead times. Nevertheless, the EC-CNN improves the overall reliability in probabilistic streamflow forecasting for all lead times as compared to EC and for most lead times as compared to EC-QM, which can benefit the downstream water resources management under uncertainty.

[Figure]

Figure 10. The CRPS of streamflow forecasts driven by the EC, EC-QM, and EC-CNN ensemble forecasts at different lead times during the test period. Error bar represents the 25th-75th percentile interval.

**References**

Bremnes, J. B. (2020). Ensemble postprocessing using quantile function regression based on neural networks and Bernstein polynomials. Monthly Weather Review, 148(1), 403-414.

Ferranti, L., Corti, S., & Janousek, M. (2018). Flow-dependent verification of the ECMWF ensemble over the Euro-Atlantic sector. Quarterly Journal of the Royal Meteorological Society, 144(712), 317-326.

**Comment:** Lien 126: What is the naive spatial resolution of the collected S2S precipitation forecasts from ECMWF?

**Reply:** The S2S precipitation reforecasts from ECMWF collected in this study are with a spatial resolution of 1.5 degrees.

The following text has been modified in the revised manuscript:

Line 142-144: The forecasted variables used in this study include precipitation, convective precipitation at the land surface, and temperature, wind components, geopotential heights, and specific humidity at 200/500/850hPa pressure levels. All of these variables are at a spatial resolution of 1.5°.

**Comment:** Line 142: EC-CNN is referenced here for the first time in the manuscript, but without a clear explanation.

**Reply:** We have added the explanation of EC, EC-CNN and EC-QM.

The following text has been added or modified in the revised manuscript:

Line 151-155: We first employ all 10 ensemble members from the ECMWF S2S gridded sub-seasonal precipitation reforecast dataset for the next 30 days as raw forecasts, denoted as EC. An ensemble of enhanced CNN models with ResNet blocks and a specialized loss function is established to statistically downscale and bias correct each ensemble member of the 1.5° EC raw precipitation forecasts to 0.25° grid resolution, with its post-processed forecast denoted EC-CNN (Section 3.2.1). The quantile mapping (QM) serves as a benchmark for comparison, with its post-processed forecast denoted EC-QM (Section 3.2.2).

**Comment:** Line 250: It seems a standardized metric is employed here (i.e., NSE) to evaluate the hydrologic model calibration. The reviewer wonders why switch to RMSE and other metrics for later streamflow predictive skill evaluation? While RMSE is a widely applied metric in many fields, standardized metrics such as NSE and KGE might be more familiar to researchers in the hydrology community.

**Reply:** In the revised version, the Nash-Sutcliffe Efficiency (NSE) is added for streamflow predictive skill evaluation.

**Comment:** Line 354: Perhaps "forecast issue date" is more appropriate for the titles of different panels in Figure 8. Also, it would be interesting to see these examples where the proposed framework delivers more accurate streamflow predictions. Overall skill evaluation would still be more informative in general. Perhaps these figures could be included in the supplementary material so that previous suggested additional evaluation and analysis could be included in the main manuscript.

**Reply:** We have renamed the titles to 'forecast issue date' in the revised manuscript. We have included the suggested analysis and evaluation in the manuscript while also keeping this figure in the revised manuscript.